# DataEnvGym: Data Generation Agents in Teacher Environments with Student Feedback

**Zaid Khan**    **Elias Stengel-Eskin**    **Jaemin Cho**    **Mohit Bansal**
UNC Chapel Hill
{zaidkhan, esteng, jmincho, mbansal}@cs.unc.edu

## Abstract

The process of creating training data to teach models is currently driven by humans, who manually analyze model weaknesses and plan how to create data that improves a student model. Recent approaches using large language models (LLMs) as annotators reduce human annotation effort, but still require humans to interpret feedback from evaluations and control the LLM to produce data the student needs. Automating this labor-intensive process by creating autonomous data generation agents – or teachers – is desirable, but requires environments that can simulate the feedback-driven, iterative, closed loop of data creation. To enable rapid and scalable testing for such agents and their modules, we introduce DataEnvGym, a testbed of *teacher environments* for data generation agents. DataEnvGym frames data generation as a sequential decision-making task, involving an agent consisting of a data generation policy (which generates a plan for creating training data) and a data generation engine (which transforms the plan into data), inside an environment that provides feedback from a student. The agent's end goal is to improve student model performance. Students are iteratively trained and evaluated on generated data, with their feedback (in the form of errors or weak skills) being reported to the agent after each iteration. As a general-purpose testbed, DataEnvGym includes multiple instantiations of teacher environments across three levels of structure in the state representation and action space, with varying levels of scaffolding support. More structured environments are based on automatically-inferred skills and offer a higher degree of interpretability and control over the curriculum. We support developing and testing data generation agents in four diverse tasks covering text, images, and actions (mathematics, programming, visual question answering, and tool-use) and test multiple student and teacher models. We find that example agents in our teaching environments can iteratively improve students across diverse tasks and settings. Moreover, we show that environments can teach different skill levels and can be used to test variants of key modules, pointing to directions of future work in improving data generation agents, engines, and feedback mechanisms. Project page: https://DataEnvGym.github.io.

## 1 Introduction

Improving an already-trained model by creating additional training data that is targeted towards current model weaknesses is an important and frequent task for researchers and engineers. For example, past work in instruction tuning and alignment has found that models can be improved with additional task-specific training examples (Touvron et al., 2023; Ding et al., 2023; Zhou et al., 2024; Chia et al., 2024; Wang et al., 2023b; Shimabucoro et al., 2024). However, the current model improvement process is largely driven by humans, who try to identify the weaknesses of the model based on evaluations, use intuition and heuristics to create data to target weaknesses, train an updated model on the data, and revise the data based on how the new model performs (Iyer et al., 2022; Longpre et al., 2023; Shao et al., 2024). The labor and repetition involved in this process strongly motivate the creation of **data generation agents** that can automate the process of creating synthetic training data to teach student models. However, there are no simulators or environments to serve as a testbed for the development of such automated teacher agents, or even to evaluate the

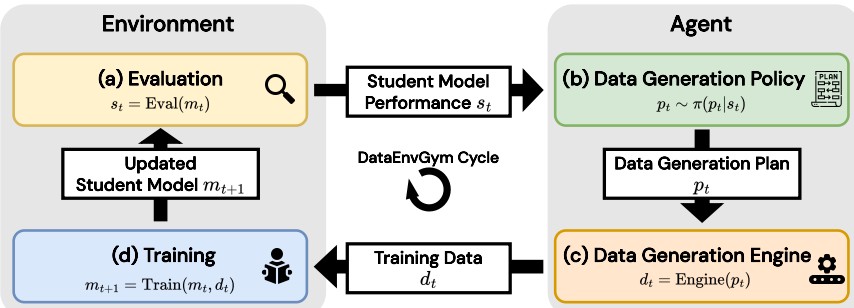

Figure 1: Overview of DATAENVGYM, a novel testbed for data generation agents. The **environment (left)** consists of evaluation (a) and training (d) of the student model. The **data generation agent (right)** takes a state encoding the current student model's performance and provides training data to improve the student model, by first creating a plan through the data generation policy (b), then executing the plan via the data generation engine (c).

effectiveness of the different components required to automate such an iterative data generation and model improvement process.

We propose DATAENVGYM, a testbed – or gym – of parameterizable teacher environments for developing autonomous data generation agents (i.e. teachers), whose goal is to iteratively improve student models by generating targeted training data conditioned on representations of the student's weaknesses (e.g. errors, inferred weak skills). We frame this task as an iterative interaction between a teacher agent and a student model in which the teacher agent creates training data that improves the student model. In the same way that game environments can be used to evaluate game-playing agents in conventional reinforcement learning (RL), DATAENVGYM's modular environments (cf. Sec. 2.1) allow us to test data generation agents for a given student model. In these environments, the data generation agent (teacher), performs multiple rounds of data creation and receives feedback from the student after each iteration in the form of student performance, which is the teacher's reward signal. We provide modules for data generation, training, and evaluation, with final performance being measured by the improvement to the student model. An overview of the DATAENVGYM can be seen in Fig. 1. First, the environment provides the agent with a *state* $s_t$, containing information about the errors of current student model $m_t$ (cf. Fig. 1(a)). Then, the agent's data generation policy $\pi$ predicts *actions* that constitute a plan for generating training data (cf. Fig. 1(b)): $p_t \sim \pi(p_t|s_t)$. Next, the agent's data generation engine executes the plan to create training data $d_t$ (cf. Fig. 1(c)). The created datapoints are then used to train an updated student model (cf. Fig. 1(d)): $m_{t+1} = \texttt{Train}(m_t, d_t)$. The updated student model is re-evaluated to produce the next iteration's state $s_{t+1}$ and provide feedback to the agent (in the form of student performance). DATAENVGYM is designed in a generalizable way to support data creation for diverse agents across multiple tasks, covering multimodal (visual question answering) and text-only (mathematics and programming) tasks.

DATAENVGYM's modular design enables many possible instantiations of data generation environments. We provide three implementations of DATAENVGYM environments along with the agent modules required for each. These differ in the state representations and action spaces they provide to the agents, and they range from open-ended (generating data directly from per-example model predictions) to more structured (generating data based on a skill-based hierarchical representation of intermediate model progress). First, in the **OPEN-ENDED environment** (cf. Fig. 2(a)), the state representation is an unstructured list of the student model's errors. The action space in this environment is also unstructured (i.e. open-ended), with an action consisting of generating a particular set of datapoints; i.e., the agent infers directly from the errors what type of data would help the model and then directly generates that data. This contrasts with human developers, who typically use a more skill-directed approach, breaking performance down into skill-specific metrics to decide where to add data. Skill-based development has three distinct advantages: it provides the agent with structured ways of controlling the data generation process, it makes the process more interpretable by organizing data generation around easy-to-grasp skills, and it enables human-model interoperability and curriculum control, where a human or a model can specify skills for the model to improve on.

Based on these advantages, we argue that skill-structured agents may be preferable or necessary in some cases. Therefore, DATAENVGYM also supports skill-based teaching and learning. Specifically, DATAENVGYM includes the **SKILL-LIST environment** (cf. Fig. 2(b)), in which a skill discovery module first automatically infers human-interpretable skills from the training data using a large language model (LLM). This produces a more structured state representation (i.e., a report of skill-specific performance), and makes the agent's task more interpretable, as it has explicit feedback on which skills the student model is struggling. Like the OPEN-ENDED environment, the SKILL-LIST environment asks agents to directly generate data. While the SKILL-LIST environment provides more structure and interpretability to the agent than the OPEN-ENDED environment by adding skills to the state representation, both have granular action spaces with a high degree of freedom. Thus, while the SKILL-LIST input space is more structured and interpretable, its output space is not. To give the agent a more structured output, we also include an environment in which the action space is structured into fixed, coarser-grained actions. To this end, in the **SKILL-TREE environment** (cf. Fig. 2(c)), we abstract skills into a tree-based representation called a **skill forest**. Here, skills are organized hierarchically into skill trees, with parent skills as root nodes and subskills as child nodes (see Fig. 3 for an example). This hierarchical framing allows new, more granular subskills to be discovered and simplifies the agent's task into a binary choice between two actions: the `explore` action, which grows a skill tree by adding new subskills, and the `exploit` action, which rebalances the skill tree to allocate more data to existing subskills. This split is designed to help the agent prioritize important skills (by generating more data for skills that have helped improve performance in the past) while also balancing competing pressures for breadth and depth in the skill hierarchy (by adding new subskills and broadening the skill tree). Our DATAENVGYM testbed not only provides default implementations for all these environments and components, but also makes it easy to test alternate implementations; for example, an improved skill discovery implementation or an alternate data structure can easily be plugged in and tested based on downstream student performance. A summary of the input and action spaces for these environments is shown in Fig. 3.

We benchmark several baseline agents as examples (data generation policies combined with data generation engines) in DATAENVGYM's teaching environments, across different domains (mathematics, programming, visual question answering) and on different student and teacher models. Generally, we find that the example agents we provide already improve student performance when models are trained in DATAENVGYM's teaching environments; after training, students see a consistent improvement when compared to their starting point (i.e., before training in the environment). Across environments, students improve by an average of $4.43\%$ (absolute accuracy) on GQA, $4.82\%$ on MATH, $1.80\%$ on LiveCodeBench, $15.17\%$ on NaturalBench, and $20.81\%$ on MnMs. Moreover, we find that our example agents can make use of student feedback to help iteratively improve the student: we compare baseline agent policies that make use of student feedback states ("With State" policies) to ones that do not ("No State" policies), finding that conditioning on the feedback state is key to successful data generation, with "With State" policies outperforming "No State" by $3.5\%$ in OPEN-ENDED, $2.05\%$ in SKILL-LIST, and $1.08\%$ in SKILL-TREE. We also show that some environments make improving students more challenging for teachers, based on how flexible versus how controllable (and interpretable) the curriculum needs to be. Moreover, we show that environments can teach different skills (based on skill frequency and question difficulty) and can be used to test variants of key modules, e.g., the skill discovery module. Lastly, we provide qualitative examples of student model predictions before and after our training. Overall, DATAENVGYM is a general-purpose testbed for developing and evaluating data generation agents, engines, and feedback mechanisms, laying the foundation for future improvements to these key elements of automated model improvement.

## 2 DATAENVGYM ENVIRONMENTS AND AGENTS

We provide three categories of (environment, agent) pairs in DATAENVGYM with multiple levels of structure given to data generation agent, corresponding to different levels of interpretability. Agents are composed of two modules: the **data generation policy** $\pi$ (which creates a data generation plan $p_t \sim \pi(p_t|s_t)$) and the **data generation engine** `Engine` (which executes the plan to produce training data $d_t = \texttt{Engine}(p_t)$; cf. Sec. 2.2.2), Both the policy and plan can change depending on the environment the agent is in, as the environment provides the agent with affordances that define the agent's action space. The environments encapsulate several modules, including a student model $m_t$, a **trainer** (which trains student model given the generated training data, $m_{t+1} = \texttt{Train}(m_t, d_t)$)

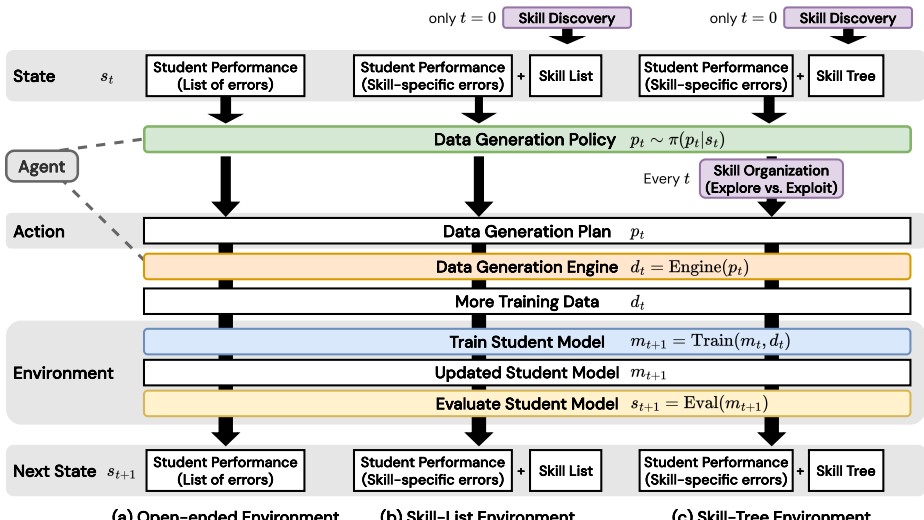

Figure 2: Illustration of the three example instances of DATAENVGYM environments described in Sec. 2. In the **(a) OPEN-ENDED environment**, the state is represented as a list of per-example accuracies, and the data generation policy directly creates a data generation plan from them. In the **(b) SKILL-LIST environment**, the state is represented as a categorized list of skills and per-skill student model performance; its data generation plan allows the policy to prioritize weak skills. In the **(c) SKILL-TREE environment**, the state is represented as a forest of *skill trees* containing skill-subskill relational information, and its data generation policy chooses between two actions for each skill: *explore* (grow skill tree) and *exploit* (rebalance skill tree).

and an **evaluator** (which evaluates the updated student model and outputs its performance, $s_{t+1} =$ Eval($m_{t+1}$); cf. Sec. 2.1.1):

As summarized in Table 1, some environments have additional modules for generating skill-specific training examples via automatic **skill discovery** (Sec. 2.1.2) and **organization** (Sec. 2.1.3). Skill-based structures give agents three distinct advantages: first, they provide the agent with affordances to control how targeted or diverse the data it generates is (i.e. knobs that adjust to what degree the data addresses a single skill vs. broadly improves a variety of skills). Secondly, when the skills are interpretable to people, skill-based agents provide *human-model interoperability and human-in-the-loop control*, where humans can influence the data generation process (e.g. by specifying skills to improve) or can in fact step in for the agent (and vice versa) whenever needed. Finally, having skill-based improvements allows for interpretable agent behavior; a user can observe for which skills data is being generated and where training is most effective. We provide all of these components for three different tasks: mathematics, visual question answering (VQA), and programming.

**(1) OPEN-ENDED Environment.** The OPEN-ENDED environment, shown in Fig. 2(a), provides the simplest state structure to the data generation agent, and is the least constrained of the environments. **State representation:** The state is represented as a list of evaluated predictions from the student. The agent must infer from this list what kind of data would best help the student, mapping directly from errors in the list to desired future datapoints. **Action space:** The action space that the OPEN-ENDED environment affords directly specifies the datapoints to generate, i.e. the agent is expected to directly generate specs for every datapoint, without any auxiliary actions to structure the generation process.

**(2) SKILL-LIST Environment.** The SKILL-LIST environment (shown in Fig. 2(b)) requires the teacher, i.e. the data generation agent, to teach a specific set of skills. **State representation:** The SKILL-LIST environment induces skills needed for the task on the training data (see Sec. 2.1.2) and reports student performance on each of these skills. The input to the agent policy is a list of evaluated predictions partitioned by skills. This informs the agent about what mistakes are being made on questions requiring the skill. **Action space:** The action space is shared with OPEN-ENDED.

**(3) SKILL-TREE Environment.** The SKILL-TREE environment, shown in Fig. 2(c), disentangles data generation from data control, adding structure to the action space s.t. its policy no longer

Table 1: Summary of baseline environments for DATAENVGYM, with different components that determine how to generate training examples for each iteration.

| Environments | Trainer/Evaluator (Sec. 2.1.1) | Skill Discovery (Sec. 2.1.2) | Skill Organization (Sec. 2.1.3) |
|---|---|---|---|
| OPEN-ENDED | ✓ | - | - |
| SKILL-LIST | ✓ | ✓ | - |
| SKILL-TREE | ✓ | ✓ | ✓ |

directly generates data specifications but simply dictates how much data is to be generated and for which subskills. This constrains the action space and provides the agent with additional scaffolding. **State representation:** The surface form of the state representation is shared with the SKILL-LIST environment. However, the SKILL-TREE environment also maintains an underlying *skill forest* composed of *skill trees*, where each tree is a hierarchical representation of a skill and its subskills (see Sec. 2.1.3 for details, see Fig. 3 for an example). Thus, while the input is similar (skill names and the student's performance on each skill) the actual skills differ from those give in the SKILL-LIST environment, which does not have any hierarchy. **Action space:** The SKILL-TREE environment affords the agent a more structured action space. At each iteration, rather than directly generating data, the agent chooses, for each skill (and its corresponding skill tree) to either `exploit` the existing skill set by *rebalancing the skill tree* for an existing skill, i.e., re-allocating the data budget to its subskills, or to `explore`, which *grows the skill tree* by creating new subskills. The action is applied to the skill tree and produces a new skill tree that has either had new subskills added or had the amount of data allocated to each subskill changed. The data generation engine then consumes each skill tree and produces the planned amount of training data for each subskill within each skill tree.

Below, we describe the constituent modules (Sec. 2.1) and the data generation agent (Sec. 2.2) that are instantiated in DATAENVGYM.

## 2.1 ENVIRONMENT MODULES

### 2.1.1 TRAINER AND EVALUATOR

Given training data $d_t$ from the data generation engine, the **trainer** performs a training run (i.e., a certain number of training steps on the dataset) updating the student model: $m_{t+1} = \text{Train}(m_t, d_t)$. Then, the **evaluator** tests the student model and outputs its performance: $s_{t+1} = \text{Eval}(m_{t+1})$. See Appendix C.1 for implementation details.

### 2.1.2 SKILL DISCOVERY

SKILL-LIST and SKILL-TREE environments have a **skill discovery** module that takes a set of training samples $d_t$ and returns a set of skills that would be needed to solve these examples; in the beginning of training $t = 0$, the environments use the skill discovery module to discover a set of skills over the validation set. Alternatively, the environments can be parameterized by a set of user-specified target skills. The skill discovery module will assign a discovered skill label to each evaluated student prediction in the validation set. The list of skills and evaluated student predictions are consumed directly by the SKILL-LIST environment. In the SKILL-TREE environment, skills are used as input by the skill organization module.

**Baseline implementation.** To discover the skills, our baseline instantiation of the skill discovery module employs a two-stage approach, following Didolkar et al. (2024). First, we assign a specific skill to each instance of a task, using a template that asks the LLM to identify the high-level skill required to solve the question. Second, we aggregate these skills into categories using another template that asks the LLM to group similar skills.

In the SKILL-TREE environment, we use the same process to propose subskills for existing skills in a hierarchical fashion. For example, given the skill "Algebra", subskills might be "Solving Linear Equations" or "Polynomial Factorization". Implementationally, we provide the LLM with a skill and existing subskills for that skill. It is instructed to propose subskills that are unique and belong to the given skill. We can repeat this process again on top of an existing skill to induce a set of subskills for each skill, as needed. The LLM prompts are shown in Appendix C.2.

Figure 3: Example skill tree updates over time for MATH task's "Algebra" skill in the SKILL-TREE environment. Starting from a empty single node, the data generation policy (Sec. 2.2.1) iteratively chooses actions between `explore` (grow skill tree) and `exploit` (rebalance skill tree). Then the skill organization module (Sec. 2.1.3) accordingly adds/removes subskills and re-allocates the training data for each subskill.

### 2.1.3 SKILL ORGANIZATION

To solve complex problems by adaptively growing the set of skills the student can perform, it is natural to organize skills into some kind of hierarchy. In the SKILL-TREE environment, the **skill organization** module takes as inputs a set of skills, and outputs a forest of "skill-trees", an organized hierarchical structure that encodes skills and stores their metadata (e.g., how much data is allocated to each skill). This is the state $s_t$ in the SKILL-TREE environment. Fig. 3 shows an example skill tree.

**Baseline implementation.** In the SKILL-TREE environment, the *skill forest* captures the student's proficiency at increasing levels of granularity, with the root of each tree corresponding to a high-level skill domain and the children corresponding to subskills. Each tree in the forest contains key information about subskills, including the amount of training data allocated to each subskill (i.e., the *data allocation*) and the student's performance on the training split for each subskill. Note that the specific implementation of the skill forest is left to the user; DATAENVGYM provides a default version of the skill forest, but other implementations can be plugged in.

### 2.2 DATA GENERATION AGENT MODULES

### 2.2.1 DATA GENERATION POLICY

The data generation policy $\pi$ takes as input the student performance state $s_t$ (list of per-example errors for OPEN-ENDED environment, skill-specific errors and the skill list for SKILL-LIST environment, and skill-specific errors and the skill tree for SKILL-TREE environment), and outputs as an action the data generation plan (the inputs to a data generation engine): $p_t \sim \pi(p_t|s_t)$. In the OPEN-ENDED and SKILL-LIST environments, the data generation plans are lists of specifications for training data, one for each training datum to be produced. The training datum is rendered or formatted into the appropriate format for instruction finetuning by the data generation engine. In the SKILL-TREE environment, we shape the action space and provide two discrete actions: explore and exploit; note that further actions can easily be added. `Explore` actions grow the skill tree by adding subskills. `Exploit` actions change the allocation of data for existing subskills.

**Baseline implementation.** We drive the policies for the OPEN-ENDED and SKILL-LIST environments with an LLM by giving the verbalized the state to the LLM and prompting it to produce the corresponding actions. For the SKILL-TREE environment, we implement a policy that grows the skill tree to a fixed size and while maintaining a uniform data distribution by sequencing explore and exploit actions. Details can be found in Appendix C.6.

### 2.2.2 DATA GENERATION ENGINE

The **data generation engine**'s role is to generate training examples based on the data generation plan from the policy: $d_t = \texttt{Engine}(p_t)$. The training examples will be used to teach the student. Because each environment affords the agent with a different action space, the data generation engines corresponding to them also differ. Specifically, for the OPEN-ENDED and SKILL-LIST environments, the data generation engine receives actions in the form of datapoints to generate (since the policy's action space is unstructured) and formats the appropriate examples (e.g. for GQA, it generates images using a T2I model). The OPEN-ENDED and SKILL-LIST generators have access to the task and a list of examples to render into training data. For the SKILL-TREE environment, where the action space is

{explore, exploit}, the data generation engine must first interpret these actions. Each action triggers a modification to the skill tree. An explore action invokes a subskill discovery pipeline to grow the skill tree by adding subskills. When emitting an exploit action, the agent has to specify how to change the data allocation, or budget, for each subskill; executing the action means adjusting the budget stored in the skill tree accordingly. Finally, the data generation engine consumes the skill tree and generates the planned amount of data for each subskill.

**Baseline implementation.** For all tasks (mathematics, VQA, and programming), we generate training data using an LLM (GPT-4o). For mathematics problems, we generate problems using an LLM, where each problem consists of a question, a step-by-step solution, and a final answer. For VQA tasks, we first use an LLM to generate image descriptions aligned with the task/skill/subskill given as input. We then employ a text-to-image model to convert these descriptions into images. Then, the LLM is instructed to generate a specified number of unique questions for the given task/skill/subskill. For programming, we generate data in two stages. We generate a problem and starter code given subskill information and detailed instructions about the expected format, and then solve it with an independent LLM call. We provide details on the generators for each environment in Appendix C.3 and show generated training examples for each task in Appendix D.

## 3 EXPERIMENTS

We experiment with DATAENVGYM environments in four domains: visual question answering, mathematics, programming and tool-use. For visual question answering, we use GQA (Hudson & Manning, 2019) and NaturalBench (Li et al., 2024); for mathematics, we use MATH (Hendrycks et al., 2021); for programming, we use LiveCodeBench (Jain et al., 2024); for tool-use, we use MnMs (Ma et al., 2024). For most experiments (reported in Sec. 3.1), we start from instruction-tuned models rather than base models because we believe it is a more realistic and challenging setting since starting from instruction-tuned models is standard for applications, and these models have undergone post-training on large amounts of task-specific data. For GQA and NaturalBench, we use PaliGemma-3b-pt-224 (Beyer et al., 2024) as our student model and we use GPT-4o OpenAI (2024) as the teacher agent policy, augmented with SDXL-Turbo (Sauer et al., 2023) for T2I generation. For MATH, we use Gemma-2-2B-Instruct (Gemma Team, 2024) as a student. For LiveCodeBench and MnMs, we use Llama-3-8B-Instruct (Llama Team, 2024) as a student. For all domains, we generate data with GPT-4o. Note that the student models we use are typically already proficient at the target task and thus are difficult to improve. For each domain, we choose the student model to satisfy the following criteria: 1) the student should be strong enough to perform the given task (e.g., LiveCodeBench is too challenging for a Gemma2-2B student). 2) The student should not have been heavily post-trained s.t. further improvements are unlikely (e.g., Llama3-8B has been extensively trained for math and further improvements with additional training are unlikely, regardless of the data generation agent). Details can be found in Appendix C.7 (validation and test splits) and Appendix C.3 (data generation). Experiments on tool-use (MnMs) and additional multimodal experiments (NaturalBench) are detailed in Appendix C.

We train all models for a fixed number of steps and terminate episodes after a fixed number of iterations, and use validation accuracy to select the training dataset iteration corresponding to the highest student accuracy. For each environment and domain, we report two values: first, we report the increase in student performance achieved by the baseline implementations of the teacher policy described in Sec. 2.2.1; this policy takes in the state $s_t$, i.e., it is given by $\pi(p_t|s_t)$. We refer to this setting as *With State*. Second, we report the increase in student performance achieved by the same policy without conditioning on the state information, i.e. sampling an action from $\pi(p_t|\cdot)$ *without* conditioning on $s_t$. For the OPEN-ENDED environment policy, we replace the list of student errors with random train samples. For the SKILL-LIST policy, we do the same. For the SKILL-TREE policy, we take explore and exploit actions with equal probability. We refer to this setting as *No State*.

### 3.1 PRIMARY RESULTS: VQA, MATHEMATICS, PROGRAMMING

Tab. 2 presents results on example instantiations of environments within DATAENVGYM. Here, we compare students before and after a multi-step trajectory of training across environments, with different data generation policies. For each setting, we report the relative gain or loss (in blue) compared to student model before training in DATAENVGYM. Note also that the models used here are already instruction-tuned on large datasets (including task-specific datasets), making obtaining further improvements particularly challenging.

Table 2: Agents in DATAENVGYM's environments are able to improve students across tasks and teaching environments. *Note that No State is the same for OPEN-ENDED and SKILL-LIST because these environments only differ in their state representation, so their No State policies (which do not condition on the state) are identical.

| Setting/Env. | GQA (PaliGemma 3B) | MATH (Gemma2 2B) | LiveCodeBench (Llama3 8B) | Avg. Improvement |
|---|---|---|---|---|
| *Before teaching* | 44.18 | 15.78 | 16.50 | - |
| OPEN-ENDED environment | | | | |
| +No State* $\pi(p_t\|\cdot)$ | 43.48 (-0.70%) | 19.78 (+4.00%) | 16.50 (-0.00%) | (+1.10%) |
| +With State $\pi(p_t\|s_t)$ | 47.90 (+3.72%) | 23.44 (+7.66%) | 18.91 (+2.41%) | (+4.60%) |
| SKILL-LIST environment | | | | |
| +No State* $\pi(p_t\|\cdot)$ | 43.48 (-0.70%) | 19.78 (+4.00%) | 16.50 (-0.00%) | (+1.10%) |
| +With State $\pi(p_t\|s_t)$ | 48.18 (+4.00%) | 19.48 (+3.70%) | 18.25 (+1.75%) | (+3.15%) |
| SKILL-TREE environment | | | | |
| +No State $\pi(p_t\|\cdot)$ | 49.53 (+5.35%) | 17.15 (+1.37%) | 16.50 (-0.00%) | (+2.24%) |
| +With State $\pi(p_t\|s_t)$ | 49.76 (+5.58%) | 18.90 (+3.12%) | 17.75 (+1.25%) | (+3.32%) |

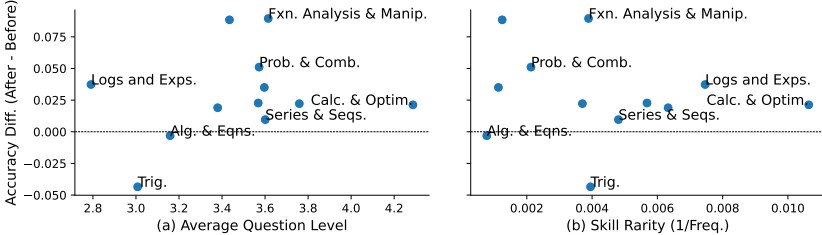

Figure 4: **Per-skill accuracy improvements** of Gemma-2B trained on MATH in the SKILL-TREE environment, as a function of **(a) question difficulty** and **(b) skill rarity (inverse of frequency)** in the training data. In both cases, the biggest performance increases occur in the middle range.

**LLM policies can make use of state information to provide better training data for the student.** Students trained in the "No State" setting generally perform worse than those trained in the "With State" setting. This is true across environments, with the largest difference (3.5%) for the OPEN-ENDED environment and the smallest difference (1.08%) for the SKILL-TREE environment. On LiveCodeBench, policies without state information are not able to improve the student at all, whereas on MATH, a policy without state information is still able to improve a student in all environments. The support provided to the teacher by the SKILL-TREE environment is particularly robust for GQA, where a policy without state information reaches almost identical performance to a policy with state information. However, absent SKILL-TREE's structured actions, removing state information actually *hurts* performance on GQA, with slight drops from the baseline for the "No State" setting on OPEN-ENDED and SKILL-LIST environments. For both these environments, "With State" improves the student model. Taken together, these results highlight the importance of the state information.

**Teaching is easier in some environments than others.** With a fixed student and task (i.e. looking at "With State" entry across the columns of Tab. 2), teachers typically elicit the highest student performance in the unconstrained OPEN-ENDED environments, where they are not required to teach a specific set of skills. However, there may be domain specific effects here as the teachers in the SKILL-TREE environment perform the best on the multimodal GQA dataset (+5.58%), whereas this is reversed for MATH, where teachers in the OPEN-ENDED environment perform the best (+7.66%).

These difference may relate to the level of constraint imposed: in the OPEN-ENDED environment, the teacher can produce any data without any constraints, while in the skill-structured environments, we require the teacher to improve the student along specified skills. This may be a more difficult task, as it may impose suboptimal constraints on the teacher, i.e., the teacher may have difficulty teaching the specified skills, whereas in the unconstrained OPEN-ENDED environment, the teacher may be implicitly able to identify and teach the skills it is best at teaching. However, unconstrained teaching (which has no clear link between errors and generated data) may not always be useful in practice, e.g. when a user wants to control the training and make targeted improvements to certain skills.

## 3.2 ANALYSIS: DIFFICULTY/RARITY, TRAINING DYNAMICS, SKILL DISCOVERY QUALITY

**Iterative training dynamics.** In Fig. 5, we plot the change in the student model's performance on the validation set throughout a full run in DATAEN­VGYM on each task and for each environment. Each experiment is truncated once the performance saturates (consistently fails to increase for multiple iterations). We use the "With State" baseline agents

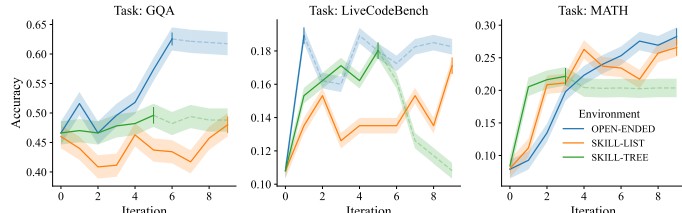

Figure 5: Training dynamics across three tasks. Each line is split into a solid segment terminating in a vertical line (the maximum performance achieved by the student) and a dashed line showing the effect of continued training beyond this point.

for each environment, and use the same models as in Tab. 2. Fig. 5 shows that the students generally improve across iterations. In other words, the baseline agents do uncover new datapoints that further improve the student at each iteration. Moreover, the behavior of agents trained in the three environments differs across tasks: on GQA, SKILL-TREE improves for more iterations than the other environments, while on MATH it reaches a plateau and on LiveCodeBench it is truncated after two rounds of data generation.

**Impact of skill discovery quality.** In Tab. 3, we show the result of training students using data generated by SKILL-LIST environments with different skill discovery modules. For each domain, we determine a set of oracle skills. For GQA, the oracle skills are the human-annotated skills. Because MATH does not have human-annotated skills, we approximate oracle skills by running the skill discovery module on the test data, thereby creating skills from privileged information. In both settings the oracle skills allow the teacher to produce better data and improve student per-

Table 3: DATAENVGYM allows us to test various implementations of environment components. Here, we compare oracle vs. inferred skills for GQA and MATH. Better skills result in better teaching and thus an improved student.

| Skill Type | GQA (PaliGemma 3B) | MATH (Gemma2 2B) |
|---|---|---|
| *Before teaching* | 44.18 | 15.78 |
| SKILL-LIST +Oracle Skills +Inferred Skills | 53.02 (+8.84%) 48.18 (+4.00%) | 19.52 (+3.74%) 18.25 (+2.47%) |

formance. The increases from oracle skills are higher for GQA than MATH, possibly due to the fact that the MATH skills still rely on the same skill discovery module as the inferred skills. This is a promising signal, as it indicates that further performance improvements can be obtained by improving the skill discovery module. These results also highlight the utility of DATAENVGYM in allowing us to swap in different components: by creating a modular framework for developing data generation agents, we enable future work to test modules and components in the framework using student performance as a metric.

**Skill learning across rarity and difficulty levels.** Tab. 2 shows that skill-based learning in the SKILL-TREE environment can improve overall performance of student models. Two core questions are (1) how interpretable these skills are and (2) how learning correlates with features like question average difficulty or skill frequency. In Fig. 4, we plot the accuracy improvement of a Gemma-2B student model after training in DATAENVGYM's SKILL-TREE environment for MATH; most skills improve, some more than others. In Fig. 4(a) we plot improvement across the average question difficulty (provided by MATH on a scale of 1 to 5). Training in DATAENVGYM boosts student performance the most in the middle difficulty region (around 3.5). On the edges, we see smaller boosts, with close to 0 difference for *Calculus and Optimization* (high difficulty) and even decreases for *Trigonometry* (low difficulty). In Fig. 4(b) we compare performance to skill rarity (inverse frequency) in the training data. Here also, the least and most represented skills benefit less. Taken together, the results in Fig. 4 suggest that there is a sweet-spot of difficulty and frequency. At the low end this could be due to saturation: easy and frequent skills benefit less from training because the model already performs well on them or has saturated. At the other end, difficult skills or very rare skills may be underrepresented in teacher's training data or be harder to generate questions for, making learning less effective. Alternatively, the questions generated may be too hard for the student. In the middle, the teacher generates helpful examples, allowing the student to learn. Similar theories have been put forth for human learning, e.g., Vygotsky (1962)'s Zone of Proximal Development,

where learning is most effective given problems slightly harder than those students could solve alone, but not so difficult that they would have no hope of solving them.

## 4 RELATED WORK

**Training Environment Generation.** In agent learning frameworks, designing training environments usually becomes a bottleneck, as it requires sophisticated human efforts (Park et al., 2024). Unsupervised environment design (UED) (Dennis et al., 2020; Jiang et al., 2021; Parker-Holder et al., 2022) explores progressively increased environment difficulty based on agent scores in simple games. Liesen et al. (2024) introduce a meta-learning approach to create learning environments for continuous control. In vision-language navigation (VLN), past work (Li et al., 2022; Li & Bansal, 2024; Wang et al., 2023c) propose augmenting the visual diversity of training environments with image generation models. Generation has been applied to game environments as well: Cobbe et al. (2020) generate a diverse set of game environments for training RL agents and measuring their generalization. Zala et al. (2024) continuously adapt game environments for training RL agents, using an LLM to generate different environments that teach core skills to the agent based on feedback from the agents' intermediate progress, which improves final performance of agents as well as learning efficiency. Yang et al. (2024) generate 3D embodied environments from user-specified prompts, generating rooms in different styles. Sudhakaran et al. (2023) and Todd et al. (2023) use LLMs for open-ended generation of game levels for tile-based games. OMNI (Zhang et al., 2023) uses LLMs for task selection in open-ended environments, while OMNI-Epic (Faldor et al., 2024) augments OMNI with the ability to generate new tasks in simulated robotics settings. Generally, past environment generation work has focused on environments like games or graphical simulations with predefined actions and skills. Moreover, past environment generation work has focused on developing *students* rather than improving the data generation process itself. In contrast, our work focuses on data generation agents, creating a testbed for teachers and treating students as part of the environment. Furthermore, our work introduces environments for data generation with automatic skill discovery in a diverse set of *open-ended and more realistic* settings such as mathematics, visual question answering, and programming.

**Learning from Generated Data.** DATAENVGYM involves transferring task knowledge from a teacher agent to a student model in an effective way, based on the student model's feedback. Past work on knowledge transfer from one model to another has been centered around knowledge distillation, where outputs from larger models are used to train smaller ones (Hinton et al., 2015; Buciluǎ et al., 2006; Chen et al., 2020); unlike the process in DATAENVGYM, this process is typically not adaptive, relying on fixed datasets of inputs that are processed by the larger teacher model and used to train the student. In the context of LLMs, symbolic distillation (West et al., 2022) has become increasingly common; here, text is generated from a large model and used to train a smaller one, e.g., in instruction tuning (Wang et al., 2023a) or distilling chain-of-thought reasoning (Wei et al., 2022) from large proprietary models into smaller models (Magister et al., 2023; Shridhar et al., 2023; Fu et al., 2023; Ho et al., 2023; Saha et al., 2023; Mukherjee et al., 2023; Mitra et al., 2023; Chen et al., 2024). The kind of teaching that data generation agents are expected to perform in DATAENVGYM's environments differs from the knowledge distillation setting in that the inputs to the model themselves are model-generated (as opposed to being sourced from an existing dataset). Moreover, the inputs are dynamically determined based on the student model's feedback and errors, whereas in knowledge distillation they are determined by a fixed dataset or generated all at once. Note that DATAENVGYM is compatible with different methods of training the student (i.e., symbolic distillation, logit-based losses, etc.), and these can be swapped into our modular environments.

## 5 CONCLUSION

We propose DATAENVGYM, a testbed of teacher environments for developing modular data generation agents (i.e., teachers) and environments. In DATAENVGYM, a teacher agent generates training data for a student model and receives feedback on the student's performance from the environment. We provide three environment instantiations with different levels of structure. Across four diverse domains (visual question answering, mathematics, programming, and tool use), we demonstrate that the example teachers we introduce can improve students in all DATAENVGYM environments. We analyze DATAENVGYM, including its training dynamics, performance across difficulty levels, and the impact of skill discovery module quality. We hope our work will foster future progress on data generation agents, engines, and feedback mechanisms by providing a testbed for evaluating and improving them.

## ETHICS STATEMENT

We experiment with improving skills of student models in visual question answering, mathematics, and programming. In our DATAENVGYM task, data generation agents are expected to create training data; we implement this process using LLMs and vision-language models, which can be biased and reflect common stereotypes as well other negative information that is present in their training data (Weidinger et al., 2021; Birhane et al., 2024). These problems impact all work seeking to generate data via models or train LLMs, and merit further exploration.

## REPRODUCIBILITY STATEMENT

We will publicly release our code and leaderboard. For all experiments, we use publicly available datasets and student models. We provide the detailed hyperparameters and additional method details in Appendix B.

## ACKNOWLEDGMENTS

This work was supported by DARPA ECOLE Program No. HR00112390060, NSF-AI Engage Institute DRL-2112635, DARPA Machine Commonsense (MCS) Grant N66001-19-2-4031, ARO Award W911NF2110220, ONR Grant N00014-23-1-2356, Microsoft Accelerate Foundation Models Research (AFMR) grant program, and a Bloomberg Data Science PhD Fellowship. The views contained in this article are those of the authors and not of the funding agency.

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

## APPENDIX

In the appendix, we present additional related work (Appendix A), additional method details (Appendix B), results on additional benchmarks (Appendix C) qualitative examples (Appendix D), experiments showing accuracy on generated data (Appendix E) and ablations on the relative importance of data vs training time (Appendix F).

## A ADDITIONAL RELATED WORK

**Skill Discovery.** A line of work in reinforcement learning has studied unsupervised skill discovery, where agents learn diverse emergent behaviors without explicit rewards. The majority of this work helps agents discover new skills by maximizing the diversity of agent trajectories, such as exploration-encouraging rewards (Gregor et al., 2016; Bellemare et al., 2016) and adding randomness during action sampling (Watkins, 1989; Burda et al., 2019). However, such methods require long exploration steps, which is expensive if the cost for agent action is not negligible. Recent work has also proposed using the knowledge contained in pretrained language models to help in skill discovery (Sharma et al., 2022; Rho et al., 2024; Fu et al., 2024) and to sample new skills (Didolkar et al., 2024). Developments and improvements to skill discovery are complementary to DATAENVGYM, where skill discovery is used in the SKILL-LIST and SKILL-TREE environments in Sec. 2.1.2) to dynamically extract human-interpretable skills from data and to create student feedback. These skills help identify model weaknesses and condition the data generation process, and we find that improving skill discovery can improve student model performance (cf. Tab. 3), pointing to directions for future work. Moreover, by using skill discovery in its environments, DATAENVGYM not only helps develop and test interpretable agents, but also provides a concrete extrinsic evaluation for skill discovery, allowing competing skill discovery methods to be evaluated and compared on the basis of how well they improve downstream agent performance.

**Model Weakness Discovery.** Testing a trained machine learning model in different scenarios is crucial in mitigating unexpected malfunctions, so that developers can actively prevent such behaviors and finetune models on weak skills. Traditionally, model weaknesses have been identified by hiring human annotators and asking them to create adversarial inputs in different scenarios and check when model outputs are incorrect or undesired (Ganguli et al., 2022). Recent work explores automatically finding model weaknesses by creating test cases in different scenarios, with either with pipelines consisting of traditional ML models (Gao et al., 2022; Wu et al., 2019) or LLMs (Perez et al., 2022; Zhang et al., 2024; Wu et al., 2024). These methods tend to focus on adversarial scenarios such as jailbreaking or redteaming, where a model is made more robust against an increasingly difficult adversary. Our framing is orthogonal: rather than focusing on defending against adversaries, in DATAENVGYM, students and teachers (data generation agents) are cooperative, working together to improve student performance.

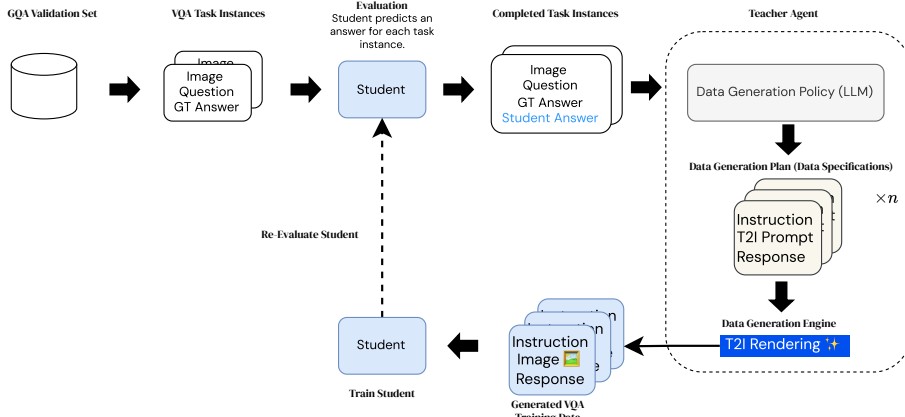

Figure 6: A complete data generation loop for the multimodal domain in the OPEN-ENDED environment. Note that the teacher agent is modular. For domains such as mathematics or code generation, a data generation engine may not be required.

## B   ADDITIONAL METHOD DETAILS

In Fig. 6, we provide a more detailed walkthrough of the core environment loop. The example in Fig. 6 depicts one experience in the OPEN-ENDED environment for the multimodal domain using the GQA dataset as the target dataset. The loop is highly modular, and can be changed to target a different data distribution by changing the dataset used for evaluation. The data generation policy and data generation engine can likewise be modified easily.

## C   ADDITIONAL MULTIMODAL AND TOOL-USE BENCHMARKS

In this section, we show results on additional challenging multimodal and tool-use tasks: Natural-Bench (Li et al., 2024) and Table 4 and MnMs (Ma et al., 2024) in Table 5. MnMs and NaturalBench include fine-grained metrics other than accuracy, which enables a more detailed view of how data generated by different methods affects a student model. On both MnMs and NaturalBench, conditioning on the task and errors of the student allows the teacher to generate more effective training data (compare "With State" against "No State" in Table 4 and Table 5).

On NaturalBench, the training data generated when conditioning on the student's state is more effective at improving the student on all metrics. The "Group Score" probes the consistency of a model across four visual question-answer pairs that test the same concept. The "Group Score" increase from generated data without conditioning on the state is zero, while conditioning on the state results in an increase in the group score.

MnMs is a tool-use benchmark that requires a modified skill discovery mechanism, as the standard pipeline applied to MnMs produces skills that are not tightly coupled to the underlying tools required. For MnMs, we consider each combination of tools as a "skill". This results in a very large number of "skills" for tool-use. To generate data on a fixed budget, we must sample a subset of skills (tool combinations) for which to generate data. We experiment with several sampling methods, and find that the most effective sampling method is to randomly sample a subset of skills for which the student does not have near-perfect accuracy.

Table 4: Performance metrics for **NaturalBench** (Li et al., 2024) before and after teaching using the OPEN-ENDED and SKILL-LIST environments.

| | Question Score | Image Score | Binary Score | Group Score | Avg. Improvement |
|---|---|---|---|---|---|
| *Before teaching* | 3.92 | 3.92 | 48.53 | 0.00 | - |
| OPEN-ENDED environment | | | | | |
| +With State $\pi(p_t\|s_t)$ | 26.00 (+22.08%) | 29.00 (+25.08%) | 61.00 (+12.47%) | 6.00 (+6.00%) | (+16.41%) |
| +No State $\pi(p_t\|\cdot)$ | 11.00 (+7.08%) | 9.00 (+5.08%) | 51.50 (+2.97%) | 0.00 (+0.00%) | (+3.78%) |
| SKILL-LIST environment | | | | | |
| +With State $\pi(p_t\|s_t)$ | 25.00 (+21.08%) | 26.00 (+22.08%) | 59.50 (+10.97%) | 2.00 (+2.00%) | (+14.03%) |
| +No State $\pi(p_t\|\cdot)$ | 11.00 (+7.08%) | 9.00 (+5.08%) | 51.50 (+2.97%) | 0.00 (+0.00%) | (+3.78%) |

Table 5: Performance metrics for **MnMs** (Ma et al., 2024) before and after teaching using the OPEN-ENDED and SKILL-LIST environments.

| | Plan Accuracy | Tool Precision | Tool Recall | Tool F1 | Avg. Gain |
|---|---|---|---|---|---|
| *Before teaching* | 37.30 | 73.74 | 68.18 | 65.41 | - |
| OPEN-ENDED environment | | | | | |
| +With State $\pi(p_t\|s_t)$ | 72.00 (+34.70%) | 87.32 (+13.98%) | 82.62 (+14.44%) | 82.25 (+16.84%) | (+19.99%) |
| +No State $\pi(p_t\|\cdot)$ | 62.13 (+24.83%) | 87.94 (+14.60%) | 87.09 (+12.91%) | 80.51 (+15.10%) | (+16.86%) |
| SKILL-LIST environment | | | | | |
| +With State $\pi(p_t\|s_t)$ | 75.96 (+38.66%) | 86.11 (+12.77%) | 84.35 (+16.17%) | 84.36 (+18.95%) | (+21.64%) |
| +No State $\pi(p_t\|\cdot)$ | 62.13 (+24.83%) | 87.94 (+14.60%) | 87.09 (+12.91%) | 80.51 (+15.10%) | (+16.86%) |

We refer the reader to the GitHub repository for complete details of prompts and hyperparameters.

## C.1 TRAINING DETAILS

We use supervised finetuning for training using the Transformers (Wolf et al., 2020) library. We present data in an instruction-finetuning format of Alpaca (Taori et al., 2023) with the standard language modeling loss. For evaluation, we use the standard training splits from the datasets we test on. More details of the training process, including hyperparameters such as learning rates and optimizers, are provided below.

### C.1.1 GQA

We use the Transformers Wolf et al. (2020) library for training. We train PaliGemma-3b-pt-224 (Beyer et al., 2024) for 10 epochs using the AdamW (Loshchilov & Hutter, 2017) optimizer with 2 warmup steps and a learning rate of $2 \times 10^{-5}$, a weight decay of $10^{-6}$ using the BF16 datatype and batch size of 16. We apply LoRA (Hu et al., 2022) with a rank of 16 and an alpha of 32, no bias, and a dropout of 0.05. We apply LoRA to all linear layers.

### C.1.2 LIVECODEBENCH AND MATH

We use Transformers (Wolf et al., 2020) and Llama-Factory (Zheng et al., 2024) libraries for training. We format all data in the Alpaca format (Taori et al., 2023) as instruction-response pairs. We use the Adam optimizer with a batch size of 16 and a cosine learning rate scheduler with a warmup ratio of 0.1 and train for 3 epochs in the FP16 datatype. We apply LoRA to all linear layers with a rank of 16 and an alpha of 32, no bias, and a dropout of 0.05. We truncate all training examples to a maximum length of 1024 tokens.

## C.2 LLM DETAILS

**LLM configuration.** We use `gpt-4o-2024-08-06` (OpenAI, 2024) for the following modules: skill discovery (Sec. 2.1.2; in SKILL-LIST env), data generation policy (Sec. 2.2.1 in SKILL-LIST env), data generation engine (Sec. 2.2.2; in OPEN-ENDED, SKILL-LIST, and SKILL-TREE envs). We use a temperature of 0 and top-p of 1.0, which are default API settings. We use the Instructor library[1] to produce structured output from LLM calls and automatically parse LLM calls into Python objects.

---

[1] https://github.com/jxnl/instructor

Table 6: Token expenditure and GPU time per iteration for each environment and domain. OPEN-ENDED environments are typically the cheapest. The most expensive domain is LiveCodeBench, although experiments in the OPEN-ENDED for code cost less than $3 per run. Training time for most environments, even with a single A6000 GPU, is typically less than 30 minutes.

| Domain | Environment | Num Tokens | Cost $ (GPT-4o-mini) | Cost $ (GPT-4o) | GPU Minutes / Iteration |
|---|---|---|---|---|---|
| Math | Open-Ended | 173234 | 0.10 | 1.73 | 24 |
| Math | Skill-List | 318528 | 0.19 | 3.19 | 24 |
| Math | Skill-Tree | 355033 | 0.21 | 3.55 | 16 |
| Coding | Open-Ended | 279304 | 0.17 | 2.79 | 16 |
| Coding | Skill-List | 497787 | 0.30 | 4.98 | 16 |
| Coding | Skill-Tree | 967610 | 0.58 | 9.68 | 16 |
| Multimodal | Open-Ended | 25073 | 0.02 | 0.25 | 37 |
| Multimodal | Skill-List | 82419 | 0.05 | 0.82 | 134 |
| Multimodal | Skill-Tree | 33991 | 0.02 | 0.34 | 78 |

**Prompt Templates.** We provide the LLM prompt templates for skill discovery (Fig. 13), data generation for GQA (Fig. 14) / MATH (Fig. 15) / LiveCodeBench (Fig. 16,), and data generation policy for OPEN-ENDED (Fig. 17) and SKILL-LIST (Fig. 18) environments.

## C.3 DATA GENERATION DETAILS

For all tasks, we use validation accuracy to identify when to terminate an episode. An episode is terminated when when a fixed number of iterations is reached, and the best student is selected from all students trained by the policy using validation accuracy.

**GQA.** We use `stabilityai/sdxl-turbo` with 4 inference steps to generate images at a resolution of $1024 \times 1024$. In the SKILL-LIST environment, our baseline policy exhausted its data budget after 3 iterations, producing 7.5k data points. In the SKILL-TREE environment, the baseline policy episode produced the top performing student after 20 steps at $\approx$ 3k datapoints. In the OPEN-ENDED environment, the baseline policy episode produced the top performing student after 5 steps and $\approx$ 3k data points.

**MATH.** For math, we follow the termination criteria as for GQA. In the SKILL-LIST environment, the baseline policy produces roughly 2.1k datapoints and the its best student after 4 iterations. In the SKILL-TREE environment, the baseline policy produces 2.8k datapoints over 20 steps to produce the best performing student. In the OPEN-ENDED environment, the baseline policy requires 10 iterations and 752 datapoints to produce its best student.

**LiveCodeBench.** For LiveCodeBench, we first generate problems, and then ask the LLM to solve the problem. Termination criteria are the same as the other settings. In the OPEN-ENDED environment, the baseline policy produces 1.6k datapoints and produces the best student after 5 iterations. In the SKILL-LIST environment, the baseline policy produces 675 datapoints and produces the best student after 11 iterations. In the SKILL-TREE environment, the baseline policy produces 3138 datapoints and produces the best student after 10 iterations.

## C.4 RESOURCE COSTS

In Tab. 6, we tabulate the token costs and GPU time required for each environment and domain. The "Coding" domain and "Multimodal" domains are most expensive with respect to token expenditure and GPU time, respectively. Moreover, we show that we can reduce the overall cost by using a different teacher model. Specifically, we conduct experiments using GPT-4o-mini (which is less expensive than GPT-4o) as the LLM powering the data generation agent, which we tabulate in Tab. 7. Here, we continue to see performance improvements even with the reduced model. In general, we expect performance improvements to remain positive but decrease as we reduce the size of the teacher model.

Table 7: Here, we replace GPT-4o inside the teacher agent with GPT-4o-mini to show that it is possible to use a less powerful but inexpensive teacher to conduct experiments.

| Setting/Env. | GQA (PaliGemma 3B) | MATH (Gemma2 2B) | LiveCodeBench (Llama3 8B) | Avg. Improvement |
|---|---|---|---|---|
| *Before teaching* | 44.18 | 15.78 | 16.50 | - |
| OPEN-ENDED environment | | | | |
| +GPT-4o-mini | 45.51 (+1.33%) | 15.78 (+0.00%) | 17.50 (+1.00%) | (+0.77%) |
| +GPT-4o | 47.90 (+3.72%) | 23.44 (+7.66%) | 18.91 (+2.41%) | (+4.60%) |
| SKILL-LIST environment | | | | |
| +GPT-4o-mini | 46.74 (+2.56%) | 16.92 (+1.14%) | 17.25 (+1.25%) | (+1.75%) |
| +GPT-4o | 48.18 (+4.00%) | 19.48 (+3.70%) | 18.25 (+1.75%) | (+3.15%) |
| SKILL-TREE environment | | | | |
| +GPT-4o-mini | 48.83 (+4.65%) | 16.06 (+0.28%) | 17.00 (+0.50%) | (+1.81%) |
| +GPT-4o | 49.76 (+5.58%) | 18.90 (+3.12%) | 17.75 (+1.25%) | (+3.32%) |

## C.5 SKILL FOREST DETAILS

The skill forest (used in SKILL-TREE environment) is a hierarchical structure representing skills and subskills across various domains. It models the student model's skill proficiency with increasing detail. Each tree in the forest corresponds to a high-level skill domain and contains the following key pieces of information for each subskill:

1. Data Allocation: The amount of training data allocated to each subskill.
2. Performance on Training Data: The student model's performance on the training data for each subskill.
3. Skill Name: The name of the skill.
4. Subskills: A list of subskills for the skill, which starts out empty.

The forest evolves through two mechanisms.

1. Growing: Adding new subskills to the tree.
2. Rebalancing: Adjusting data allocation for existing subskills based on performance.

These actions conceptually correspond to exploring and exploiting the skill tree.

This structure allows us to represent fine-grained skills, allocate resources for data generation, track the student model's progress, and prioritize skills in the data production process.

## C.6 SKILL-TREE POLICY

We develop a policy as the baseline "With State" policy shown in Tab. 2. that aims to grow and balance a skill tree. It operates in two phases: Growth Phase: The policy alternates between exploration and exploitation actions until a predetermined maximum number of subskills is reached. During exploration, new subskills are added to the tree. During exploitation, the policy resets data allocations to zero, preparing for the next exploration. Filling Phase: Once the maximum number of subskills is reached, the policy switches to a pure exploitation strategy. It calculates and executes actions that incrementally allocate data to each subskill, aiming to reach a specified maximum amount of data per subskill. The policy respects a maximum allocation limit per action and continues until all subskills have reached their data capacity.

## C.7 VALIDATION AND TEST SPLITS

For GQA, we create validation and test splits by doing a balanced stratified sampling of the validation and testdev sets repeatedly. Specifically, we sample 5 questions from each of the 100 question types in GQA, following Gupta & Kembhavi (2023). For MATH, we create a validation set by doing balanced stratified sampling of the test set across all levels and topics in MATH, selecting 50 from

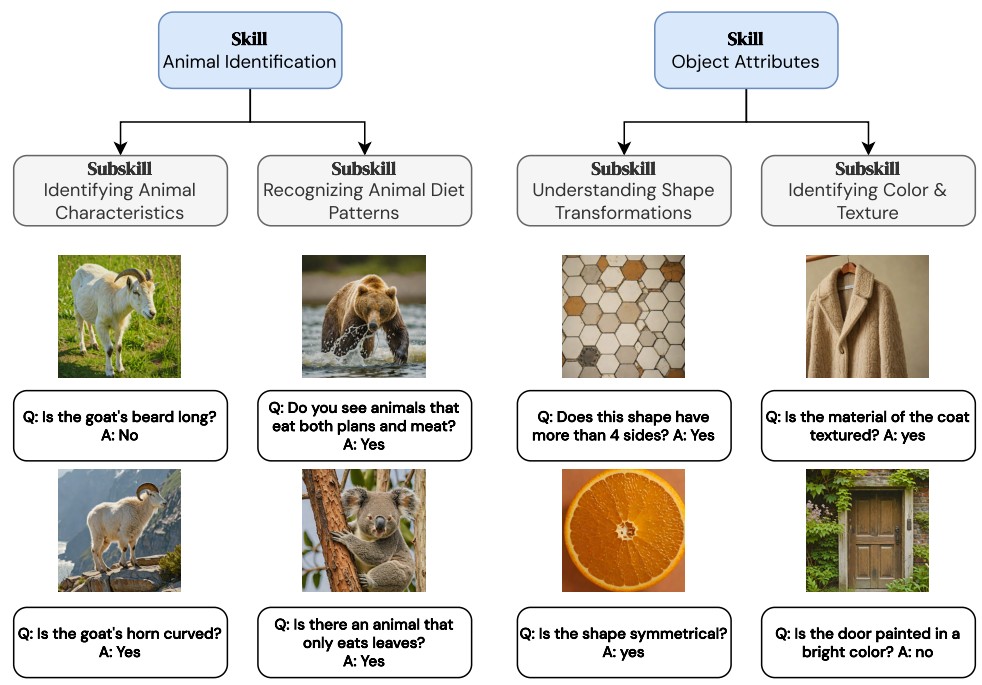

Figure 7: A partial view of a skill forest for GQA. Depicted are 2 out of 13 discovered skills. For each skill in the skill tree, we show 2 subskills and 2 examples of generated data for that subskill. Note that all images are generated.

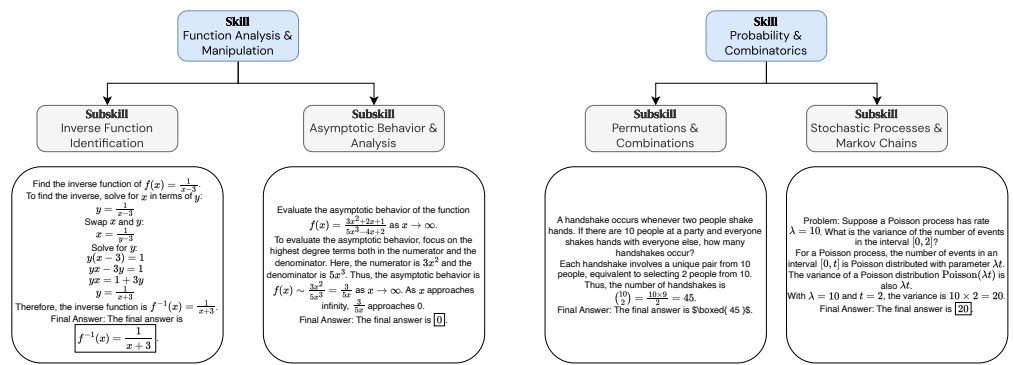

Figure 8: A partial view of a skill forest for MATH. Depicted are 2 out of 12 discovered skills. For each skill in the skill tree, we show 2 subskills and 1 example of generated data for that subskill.

each group. We use the official test set for MATH. For LiveCodeBench, we create a validation set by choosing all problems that are in the 2nd release but not in the 1st release as our validation set, and use the entire 1st release as our test set. This results in a relatively small validation set of only 100 problems.

## D    QUALITATIVE EXAMPLES

**Generated skill trees and examples.**    In Fig. 7, Fig. 8, and Fig. 9 we show qualitative examples of skills and subskills discovered for GQA, MATH, and LiveCodeBench, respectively. For each subskill,

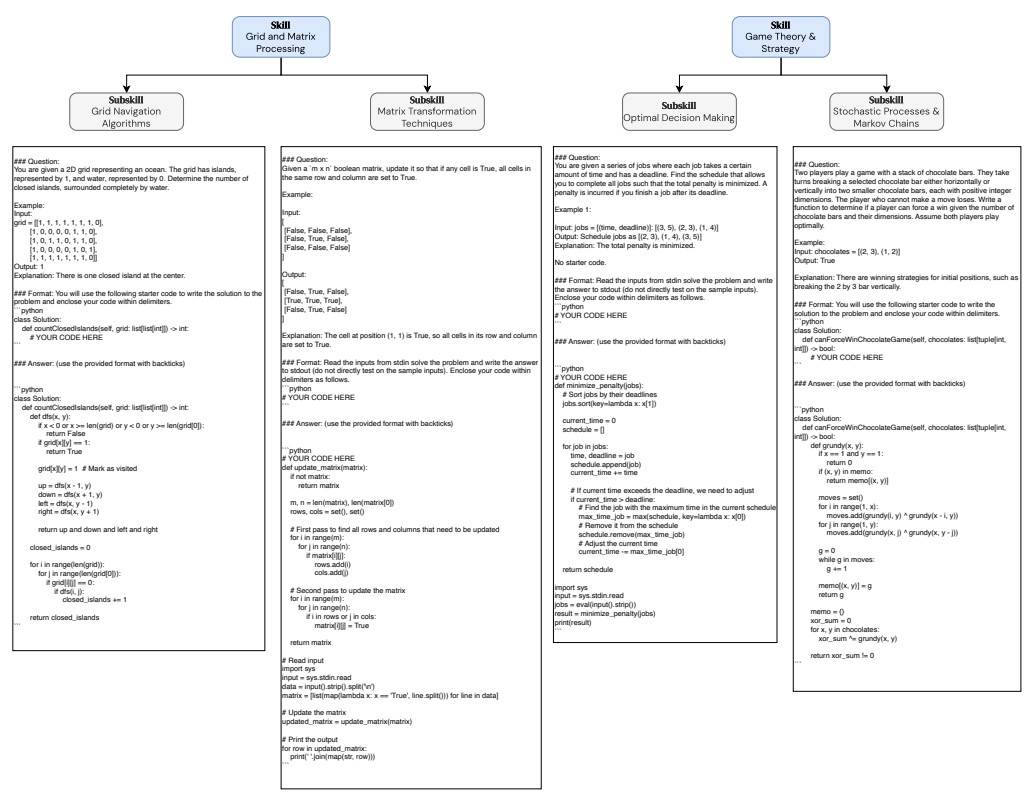

Figure 9: A partial view of a skill forest for LiveCodeBench. Depicted are 2 out of 15 discovered skills. For each skill in the skill tree, we show 2 subskills and 1 examples of generated data for that subskill.

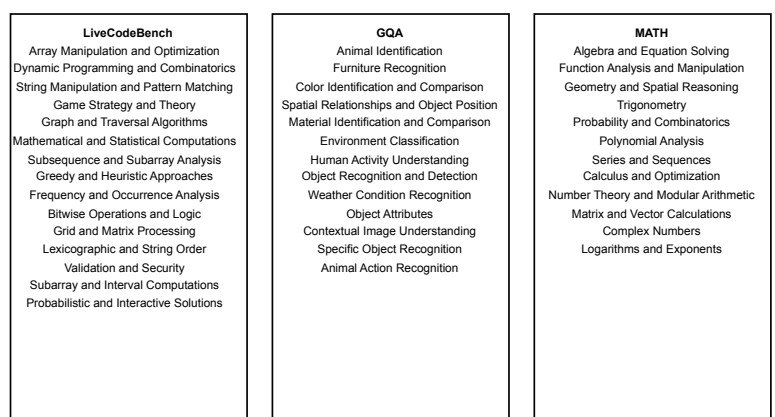

Figure 10: Skill lists for each domain. These were determined by the skill discovery module and used in the SKILL-TREE and SKILL-LIST environments.

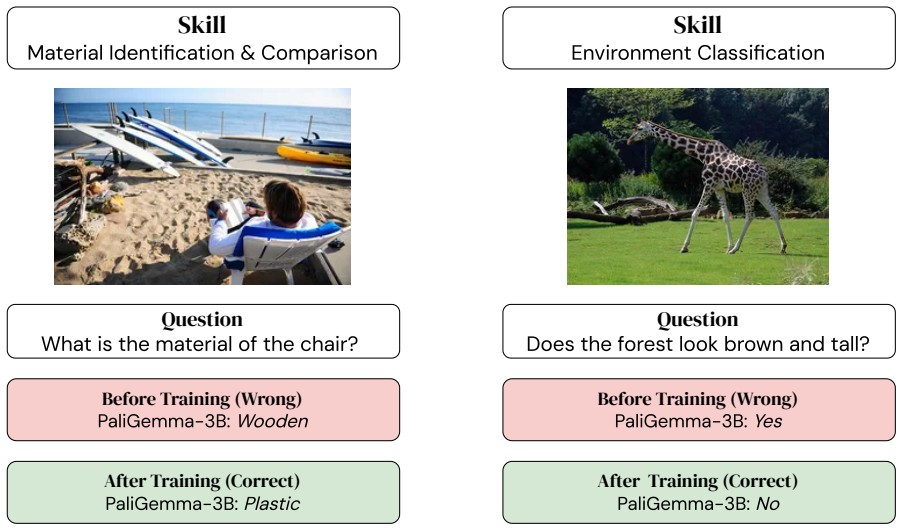

Figure 11: Qualitative examples of how training on generated data changes the response of a PaliGemma-3B student.

we also show example datapoints generated by the teacher model. Note that these datapoints are generated in their entirety, including the images shown in Fig. 7.

**Model predictions before and after training.** In Fig. 11 we show qualitative examples of how training on generated data changes the response of a PaliGemma-3B student. The example on the left falls under the "Material Identification and Comparison" skill that was identified during skill discovery. Training on generated data leads to the model correctly identifying the material as "plastic". This may be a result of debiasing in terms of the possible materials for chairs in the synthetic data. On the right is another tricky example, where the initial answer could be a result of the model confusing the foreground (a giraffe) – for which "brown and tall" is true – with the background. After training on synthetic data, this mistake is corrected.

## E   ACCURACY ON GENERATED DATA

We analyze the performance of the student on training data generated by the teacher over successive iterations. We plot this data in Fig. 12. Concretely, for the Math and Multimodal environments, we plot the accuracy of a student at iteration $n$ on data generated in iteration $n + 1$; thus, the data is generated but it has not been seen by the student yet. The student improves on data generated by the teacher through successive iterations as seen in Fig. 12. This shows that the data produced by the teacher is consistent over iterations and the student is gradually improving at the target data distribution. However, the data distribution produced by the teacher is complex enough that the improvement is neither monotonic or accomplished in a single iteration.

## F   ABLATION: DATA VS TRAINING

To substantiate that claim student performance is increased due to added data points rather than insufficient training, we take a subset of the data and increase the number of epochs such that the student receives a fraction of the added data, but an equivalent number of epochs as training on the full data. For example, if a student is normally trained for 10 epochs with 1000 generated training datapoints, we take the data from the first data generation iteration (which might, for example, contain 200 training datapoints) and train an alternative student for $\frac{1000}{200} \times 10 = 50$ epochs to isolate the effect of the generated training data vs the added training epochs. We show the results in Tab. 8. In each case,

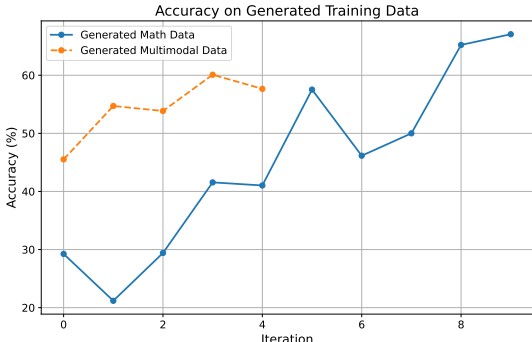

Figure 12: Over successive iterations, the student becomes more proficient at data generated by the teacher. Here, we plot the accuracy of a student from iteration $n$ on data generated by the teacher in iteration $n + 1$, which has not been seen by the student.

| | Mutimodal (GQA) | | | MATH | | | Coding (LiveCodeBench) | | |
| --- | --- | --- | --- | --- | --- | --- | --- | --- | --- |
| | Data | Epochs | Accuracy | Data | Epochs | Accuracy | Data | Epochs | Accuracy |
| Before Teaching | - | - | 44.18 | - | - | 15.78 | - | - | 16.50 |
| Less Data / Longer Training | 20% | 15 | 42.79 | 10% | 30 | 13.98 | 20% | 15 | 15.00 |
| More Data / Standard Training | 100% | 3 | **47.90** | 100% | 3 | **23.44** | 100% | 3 | **18.91** |

Table 8: Training with less data but for more epochs produces significantly smaller improvements than training with more data for fewer epochs, showing that data is responsible for increased performance rather than more training.

training with less data but for more epochs produces substantially smaller improvements than training with more data for fewer epochs, showing that data is responsible for increased performance rather than more training. In fact, extending training without additional data typically hurts performance — fresh data is essential. This highlights the importance of studying data generation as we do, as data generation is one of the few ways of obtaining fresh data.

```
QUESTION_TO_SKILL_TEMPLATE = jinja2.Template(
    """Consider this question about {{ topic }}.
The question tests the {{ topic }} skills of a {{ student_description }}.
Label this question with a specific skill that would be required to solve the question.
You should be able to use the skill as a dictionary key in python.
The skill name should be lower case letters only.
The skill name should be very descriptive and you may use multiple words to describe the skills required in the question.
If you do use multiple words per question, then join them by an underscore.

Question: {{question}}
"""
)

SKILL_TO_CATEGORIES_TEMPLATE = jinja2.Template(
    """Here is a list of skills required by a {{ student_description }} to solve questions about {{ topic }}:
{% for skill in skills %}
- {{- skill -}}
{% endfor %}

Reduce the number of unique skills by grouping similar skills into categories and give a descriptive name to each category.
When choosing categories, consider the following:
- The categories should be mutually exclusive.
- The categories should be collectively exhaustive.
- The categories should be descriptive of the skills they contain.

When designing the skill categories, keep in mind that we want to use the skill categories to guide training data collection to improve a {{ student_descriptor
}}'s performance on {{ topic }} tasks.
Design the skill categories so that collecting data for each category will help improve the model's performance on the underlying skills.

{% if num_categories is not none %}
Group the skills into at least {{ num_categories }} categories.
{% endif %}
{% if no_function_calling %}
Respond with a list of lines, where each line is a category name followed by a colon and a list of representative skills for that category.
Here is a Python template demonstrating the expected format:
```python3
template = "{index}. {skill_category}: {representative_skills}"
```
Here are examples:
1. Bird Identification: identifying_birds, recognizing_birds, bird_species
2. Welding: welding_steel, welding_aluminum, welding_titanium

A skill category may encompass as many skill as you think are appropriate, but only list up to 3 representative skills.
Produce plain text, do not wrap the your response in backticks or triple backticks.
Write nothing but the lines that follow the template.
{% endif %}
"""
)

LABEL_QUESTIONS_WITH_SKILL_CATEGORIES_PROMPT = jinja2.Template(
    """Your task is to identify the skill required to solve a question that tests {{ topic }}.
Here is a list of possible skills required by the question:
{% for skill in skills %}
- {{- skill -}}
{% endfor %}
Label the question with one skill from the list, and provide a reason for your choice.

You must ALWAYS choose a skill from the list of skills provided.

Question: {{question}}
"""
)
```

Figure 13: LLM prompt templates for skill discovery.

```
You are a experienced machine learning engineer and your role is create training data for a model.

Here are some examples of the style of question the model will be answering, and the correct response to the question:
{% for example in examples %}
- instruction: {{ example.instruction }}
- response: {{ example.ground_truth_label }}
{% endfor %}

We will focus on improving skills underneath the category of "{{ subskill }}".

You will propose hypotheses about what training data the model needs to improve its skills under "{{ subskill }}".
The hypotheses will contain specifications of the training data, and we will generate the data from those specifications, and then train the model on the data.

The training data you produce must be valid JSON using the provided schema.
Here are descriptions of the fields in the schema:
- "instruction": The instruction you want the model to respond to.
- "image": The description of an image the instruction is about.
- "response": The correct response to the instruction.

When crafting the training data, consider the following:
- the instructions should be similar in style, length, and complexity to the examples provided
- the images should be relevant to the instruction
- the responses should be similar in style, length, and complexity to the examples provided
- think about what knowledge the model might be missing that would help it answer the question correctly, and craft your training data to give it that knowledge
- each response should be a logically _correct_ response to the instruction in the context of the image description
- the training data should be diverse and help the model improve on "{{ subskill }}"

Produce {{ num_data_specs }} specifications for training data.
```

Figure 14: LLM prompt template for data generation - GQA.

```
You are an experienced math educator and your task is to create math problems for improving a student's skills in solving math problems.

{% if already_generated_data %}
Here are some problems that you have already written:
{% for data in already_generated_data %}
- Problem: {{ data.problem }}
  - Chain of Thought: {{ data.chain_of_thought }}
  - Final Answer: {{ data.final_answer }}
{% endfor %}
{% endif %}

Each problem should improve the student's ability to solve problems under the category of "{{ subskill }}".
Each problem should require the student to know the concept of "{{ subskill }}".
The problems you produce must be valid JSON using the provided schema.
Here are descriptions of the fields in the schema:
- "problem": The math problem you want the model to solve. Ensure this is valid LaTeX that is properly escaped for representation as a string in Python.
- "chain_of_thought": A step-by-step explanation of how to solve the problem. Ensure this is valid LaTeX that is properly escaped for representation as a string in Python.
- "final_answer": The final answer to the problem as a LaTeX string. For example '17' or '\frac{1}{2}' or `\\matrix{1 & 2 \\cr 3 & 4}`. Do not write a sentence here, just the answer.

Propose {{ data_budget }} new problems.
```

Figure 15: LLM prompt template for data generation - MATH.

```
You are an expert Python engineer and competitive programming tutor.
You are helping a junior engineer improve their coding skills.

{% if lcb_examples %}
Here are representative examples of the kind of coding problems the junior engineer is facing.
{% for example in lcb_examples %}
Problem:
{{ example.instruction }}
{% if example.starter_code %}
Starter Code:
{{ example.starter_code }}
{% endif %}
{% if example.solution %}
Solution:
{{ example.solution }}
{% endif %}
{% endfor %}
{% endif %}

You are focusing on problems requiring skills in the category of "{{ subskill }}".

You will propose a new set of problems that require applying skills in the category of "{{ subskill }}".
The problems you propose should be such that solving them will help the junior engineer improve their skills in the category of "{{ subskill }}".

Here are some guidelines:
- the problems should be similar to coding problems on platforms like LeetCode, Codeforces, etc.
- the problems should require applying skills in "{{ subskill }}"
- only propose problems that YOU KNOW the solution to. This is CRITICAL.

# Output Format
For each problem, you need to include the following:
- instruction: A complete problem statement that would be found in a place like LeetCode. This will be shown verbatim to the junior engineer.
    - This should include an example input / output and a concise explanation for why the output is correct.
    - Do not write "### Question", just output the problem statement.
- starter_code: The starter code to the problem. Not all problems need starter code.

If you are including starter code, it should be formatted as follows:
```python
class Solution:
    def functionWithMeaningfulName(self, parameter_1: list[SomeType], parameter_2: AnotherType):
        # YOUR CODE HERE
```

Keep "# YOUR CODE HERE" in the code block so the junior engineer knows where to fill in the solution.
You can change functionWithMeaningfulName to anything you want.
Don't forget to also change the parameter names to something that makes sense for the problem.

Propose no more than {{ num_data_specs }} new problems.
{{ num_no_starter_code_problems }} should have no starter code.
The remaining {{ num_data_specs - num_no_starter_code_problems }} should have starter code.
```

Figure 16: LLM prompt template for data generation - LiveCodeBench.

```
You are a experienced engineer and your role is to provide training data to correct a model.

The model was given the following instruction and responded incorrectly.
Instruction: {{ vqa_task_error.task_instance.instruction }}
Model Response: {{ vqa_task_error.predictor_response }}
Correct Response: {{ vqa_task_error.task_instance.ground_truth_label }}

Craft training data to improve the model. The model will be trained on the data you provide.
The training data you produce must be valid JSON with the following fields:
- "inferred_weak_skill": A concise to-the-point description of why you think the model responded incorrectly and how you'll fix it. Produce this first to give yourself a chance to think.
- "instruction": The instruction you want the model to respond to.
- "image": The description of an image the instruction is about.
- "response": The correct response to the instruction.

When crafting the training data, consider the following:
- the instructions should be similar in style, length, and complexity to the original instruction
- the images should be relevant to the instruction
- the responses should be similar in style, length, and complexity to the original response
- think about what knowledge the model might be missing that would help it answer the question correctly, and craft your training data to give it that knowledge
- each response should be a logically _correct_ response to the instruction in the context of the image description
- the training data should be diverse enough to help the model generalize to new examples

Produce no more than {{ num_training_data }} training data examples.
```

Figure 17: LLM prompt template for generation policy for OPEN-ENDED environment.

```
You are an experienced math educator and your task is to create training data for improving a model's skills in solving math problems, especially under the category
of "{{ skill_category }}".

Here are examples of mistakes the model made while solving problems requiring "{{ skill_category }}".
The model was given a math problem and responded incorrectly.
{% for math_task_error in math_task_errors %}
- Problem: {{ math_task_error.task_instance.instruction }}
  - Model Response: {{ math_task_error.predictor_response }}
  - Correct Response: {{ math_task_error.task_instance.ground_truth_label }}
{% endfor %}

You will propose hypotheses about what training data the model needs to improve its skills under "{{ skill_category }}".
For certain skills, the model may not have made any mistakes. In that case, propose hypotheses that will help the model improve on harder examples of the skill.

The training data you produce must be valid JSON using the provided schema.
Here are descriptions of the fields in the schema:
- "inferred_weak_skill": A concise description of the skill under "{{ skill_category }}" that the model is weak at, and what kind of (problem, response) data will help the
model improve.
- "problem": The math problem you want the model to solve. Ensure this is valid LaTeX that is properly escaped for representation as a string in Python.
- "chain_of_thought": A step-by-step explanation of how the model should solve the problem. Ensure this is valid LaTeX that is properly escaped for representation as
a string in Python.
- "final_answer": The final answer to the problem as a LaTeX string. For example '17' or '\\frac{1}{2} or `\\matrix{1 & 2 \\cr 3 & 4}`. Do not write a sentence here, just
the answer.

Produce {{ num_hypotheses }} hypotheses.
For each hypothesis and weak skill, produce {{ num_data_specs }} specifications for training data.
```

Figure 18: Example of a prompt for the SKILL-LIST environment for MATH.

