# OpenReview forum: "DataEnvGym: Data Generation Agents in Teacher Environments with Student Feedback"
_ICLR.cc/2025/Conference — ICLR 2025 Spotlight_

### Official Review · Reviewer_wuGW · 2024-11-01

**Soundness:** 3
**Presentation:** 2
**Contribution:** 3
**Rating:** 8
**Confidence:** 4

**Summary:**

This paper introduces DataEnvGym, a novel testbed of teacher environments for developing data generation agents that iteratively improve student models by generating targeted training data. DataEnvGym frames data generation as a sequential decision-making task where an agent, comprising a data generation policy and engine, interacts with an environment that provides feedback from a student model. The agent's goal is to improve student model performance by generating training data based on student feedback (errors or weak skills). DataEnvGym offers multiple instantiations of teacher environments across three levels of structure: open-ended, skill-list, and skill-tree, each with varying levels of scaffolding support. Experiments across text and image-based tasks (mathematics, programming, and visual question answering) demonstrate that example agents within DataEnvGym can iteratively improve student model performance. Furthermore, the authors analyze the impact of state information, environment structure, and skill discovery quality on agent performance and student learning. The paper concludes that DataEnvGym, with its modular design and support for diverse tasks and student models, provides a valuable platform for developing and evaluating data generation agents, engines, and feedback mechanisms for automated model improvement. The code and leaderboard will be publicly released.

**Strengths:**

Novel Problem: Automating data generation to improve models is a significant challenge with practical applications. This work directly addresses this problem with a novel approach.

Well-Defined Framework: DataEnvGym is presented as a well-defined framework with clear components (trainer, evaluator, data generation policy, data generation engine) and different levels of structure (open-ended, skill-list, skill-tree). This structure makes the problem tractable and facilitates modular development and testing.

Multiple Tasks and Domains: The inclusion of experiments across diverse tasks (mathematics, programming, visual question answering) and with different student models demonstrates the generalizability of the framework.

Promising Results: The initial results showing improved student model performance across tasks and environments are encouraging and suggest the potential of this approach. The analysis of difficulty/rarity and training dynamics adds value.

Open-Source Release: The commitment to publicly releasing the code and leaderboard promotes reproducibility and encourages further research in this area.

**Weaknesses:**

Limited Evaluation of Agent Architectures: The focus is primarily on the environment itself, with less emphasis on the architecture and training of the data generation agents. While baseline agents are provided, more sophisticated agent designs (e.g., reinforcement learning agents, agents leveraging larger language models) and their systematic evaluation would significantly strengthen the paper. How do different agent architectures compare in terms of effectiveness and efficiency? Are there specific architectural choices that are particularly well-suited for this task?

Over-Reliance on LLMs for Data Generation: While using LLMs for data generation is a reasonable starting point, it raises concerns about the quality and diversity of the generated data. Exploring alternative data generation methods (e.g., data augmentation techniques, programmatic data generation) and comparing their effectiveness with LLM-based generation would be valuable. How robust is the framework to the quality of the generated data?

Limited Analysis of Skill Discovery Quality: While the paper briefly touches upon the impact of skill discovery quality, a more thorough investigation is needed. How does the quality of the discovered skills affect the performance of the data generation agents and the student models? What are the limitations of the current skill discovery method, and how can it be improved? Quantitative analysis of skill quality (e.g., measuring coherence, coverage, and relevance) would strengthen the paper.

Lack of Comparison with Existing Methods: While related work on knowledge distillation and model weakness discovery is discussed, there's no direct comparison with existing methods for model improvement. How does DataEnvGym compare to techniques like curriculum learning or active learning in terms of effectiveness and efficiency? Including such comparisons would better contextualize the contributions and highlight the advantages of the proposed approach.

Limited Discussion of Scalability: The experiments are conducted with relatively small datasets and models. How does DataEnvGym scale to larger datasets and more complex models? What are the computational challenges associated with training data generation agents in more realistic settings? Addressing these scalability concerns is crucial for practical applications.

**Questions:**

Limited Evaluation of Agent Architectures: The paper primarily focuses on introducing the DataEnvGym environment, but the evaluation of data generation agents is limited to relatively simple baseline policies. Exploring more sophisticated agent architectures, such as reinforcement learning agents (e.g., using policy gradient methods, Q-learning) or agents incorporating larger language models for planning and decision-making (similar to the approaches used in Shimabucoro et al. (2024), would substantially strengthen the paper. A systematic comparison of different agent architectures in terms of their effectiveness in improving student models, their sample efficiency, and their computational cost would provide valuable insights and contribute to a better understanding of the challenges and opportunities in automated data generation.

Limited Analysis of Skill Discovery Quality: The paper briefly discusses the impact of oracle skills on student performance but doesn't delve deeply into the quality of the skills discovered by the proposed LLM-based method. A more thorough analysis is needed to understand the strengths and limitations of the skill discovery module. This could involve quantitative measures of skill quality, such as measuring their coherence, coverage, and relevance to the target task, or qualitative analysis by human experts. Investigating how the quality of the discovered skills affects the performance of the data generation agents and the resulting student models would strengthen the paper's contribution. Exploring alternative skill discovery methods (e.g., clustering-based approaches, topic modeling) and comparing their effectiveness with the proposed method would further enhance the analysis.

Lack of Comparison with Existing Methods: The paper positions DataEnvGym as a novel approach for model improvement, but it lacks a direct comparison with existing methods like curriculum learning (Bengio et al., 2009) or active learning (Settles, 2009). Evaluating how DataEnvGym compares to these established techniques in terms of student model performance, data efficiency, and computational cost would provide valuable context and highlight the advantages of the proposed framework. This would also clarify the specific niche and contribution of DataEnvGym within the broader landscape of model improvement techniques.

Limited Discussion of Scalability: The experiments in the paper are conducted with relatively small datasets and models. It's essential to address the scalability of DataEnvGym to more realistic scenarios involving larger datasets, more complex models, and a broader range of skills. Discussing the computational challenges and potential optimizations for scaling the framework to more demanding settings would strengthen the paper's practical relevance. For instance, how can the computational cost of LLM-based data generation be reduced while maintaining data quality? How can the skill discovery and agent training processes be optimized for larger datasets? Addressing these questions would provide valuable insights for future research and practical applications.

---

> ### Author Response · Authors · 2024-11-20
> **Response to Reviewer wuGW**
>
> Thank you for recognizing the value of DataEnvGym and pointing out the “_novel problem_” we address as well as our “_well-defined framework_” and “_promising results_”.
>
> **W1/Q1: We include multiple agent architectures and experiment with two additional teacher LLMs**.
> Each environment requires a different agent architecture, so we have 3 in total. We also try with different teacher LLMs: GPT-4o (Table 2, L378-393) and GPT-4o-mini (Table 5, L972-986 in the revised PDF).
>
> **W2: Data is generated by several components working together.**
> In addition to generating data via LLM, we experiment with multimodal grounding datasets.
> In these cases, the data is generated by a text-to-image model. In all cases, the LLM is only a component of a pipeline that involves many modules, such as skill discovery and a data generation engine. For example, in the SKILL-TREE environment, the policy making decisions about what skills to generate data for are not LLMs and can be classical controllers.
>
> **W3/Q2: We have added additional analysis of the learned skills.**
> Following your suggestion, we have added an additional figure showing a full list (Figure 10, L1115-1132 in the revised PDF) of discovered skills for MATH, GQA, and LiveCodeBench in the SKILL-LIST environments in Appendix C. We have also added another figure showing qualitative examples of skill-errors that were fixed by training on synthetic data in Appendix C, highlighting the utility of skills in our framework. In summary, we now have 5 figures showing qualitative examples of skill discovery and one quantitative analysis of skill discovery in Appendix C.
>
> **W4/Q3: We compare with active learning.**
> We implement data selection using prototypicality scores \[A\] which are standard for active learning. Similar to the random selection baseline, it is hard to improve a well-post-trained LLM like Llama3 or Gemma2 by using readily available data pools — it is much easier to improve them using generated data. Even using the full training dataset cannot improve the student. This motivates our choice to tackle data generation rather than data selection. The training of open-source frontier models (Llama3, for example) includes significant post-training that subsumes publicly available data sources \[B, $\\S$4.2, C$\\S$4\], making it hard to improve them with any amount of already existing data.
>
> |  | Before Training | Data Selection (Prototypicality) | Full Training Dataset | Data Generation (Open-Ended) |
> |---|---|-----|---|----|
> | MATH Accuracy | 15.78  | 16.01  | 15.18  | **23.44**  |
>
> **W5/Q4: DataEnvGym has been designed for scalability.**
> On a single A6000, the total training time for our most computationally expensive setting (multimodal) is 6h, or about 1.5h/iteration. Most other settings are much faster. Environments are fully parallelizable using Ray and can be scaled up to multiple GPUs and even multiple nodes. We’ve added a full accounting of token and GPU costs in Table 4, L918-931 of the revised PDF.
>
> \[A\] Sorscher et al., Beyond neural scaling laws: beating power law scaling via data pruning, NeurIPS 2022 Outstanding Paper Award
> \[B\] Llama Team, AI @ Meta, The Llama 3 Herd of Models, arXiv 2024
> \[C\] Gemma Team, Google Deepmind, Gemma 2: Improving Open Language Models at a Practical Size, arXiv 2024

---

> > ### Comment · Reviewer_wuGW · 2024-11-27
> >
> > Thank you so much for looking into my feedback and working on it. I am in the process of reviewing the updated manuscript and will let you know but so far you have pretty much addressed my concerns. Cheers!

---

> > > ### Author Response · Authors · 2024-11-27
> > >
> > > Thank you Reviewer wuGW for your feedback/engagement and positive appraisal of our work! We're glad our rebuttal was able to address your questions.

---

### Official Review · Reviewer_rVo8 · 2024-11-04

**Soundness:** 2
**Presentation:** 3
**Contribution:** 3
**Rating:** 6
**Confidence:** 4

**Summary:**

The paper presents DataEnvGym, a framework designed to simulate environments for data generation agents. These agents iteratively generate synthetic data to address weaknesses in student models, aiming to improve model performance across tasks like mathematics, programming, and visual question answering. DataEnvGym provides various structured environments (Open-Ended, Skill-List, and Skill-Tree) where data generation agents create targeted training examples based on feedback from the student model, offering a dynamic approach to automated model improvement.

**Strengths:**

- Good contribution to automated data generation for model improvement.
- Clearly written with structured sections explaining each environment type and experimental results.

**Weaknesses:**

- The paper should clarify early on that the focus is on synthetic data generation for training purposes, as this underpins the motivation for the approach.
- Important related works on algorithms using feedback from training to generate the next training environments are missing [1, 2, 3, 4].
- Lines 460 - 465, I believe there is a typo whereby it says that “each experiment is truncated once the performance consistently decreases for multiple iterations”. Should it be “increases”?
- Repeated runs of experiments without confidence intervals will be valuable, especially since the variance of performance seems to be very high.

[1] Sudhakaran, S., González-Duque, M., Freiberger, M., Glanois, C., Najarro, E., & Risi, S. (2024). Mariogpt: Open-ended text2level generation through large language models. Advances in Neural Information Processing Systems, 36.
[2] Todd, G., Earle, S., Nasir, M. U., Green, M. C., & Togelius, J. (2023, April). Level generation through large language models. In Proceedings of the 18th International Conference on the Foundations of Digital Games (pp. 1-8).
[3] Zhang, J., Lehman, J., Stanley, K., & Clune, J. (2023). Omni: Open-endedness via models of human notions of interestingness. arXiv preprint arXiv:2306.01711.
[4] Faldor, M., Zhang, J., Cully, A., & Clune, J. (2024). OMNI-EPIC: Open-endedness via Models of human Notions of Interestingness with Environments Programmed in Code. arXiv preprint arXiv:2405.15568.

**Questions:**

- How does the performance of the data generation agents change over longer iterations? The paper truncates experiments when performance increases, but it would be insightful to explore whether performance plateaus or continuously increase over extended training.
- Is the total training data allocation fixed in each environment, or does it vary dynamically? The methodology mentions rebalancing but lacks clarity on how these allocations adjust adaptively based on feedback.

---

> ### Author Response · Authors · 2024-11-20
> **Response to Reviewer rVo8**
>
> Thank you for stating that we make a good contribution to automated data generation and quality feedback\!
>
> **W1: We clarify that our focus is on synthetic data generation for training purposes**.
> We have added and highlighted text to the introduction in L050-051 in the revised PDF that clarifies our focus is on data generation for training purposes.
>
> **W2: Related works.**
> Thanks for providing the additional related works that fit into our section focused on simulations/games with a fixed set of actions and skills. We have cited them and discussed them in Section 4 under the paragraph “Training Environment Generation” (L503-506 in the revised PDF).
>
> **W3: We truncate experiments when performance decreases**.
> This is not a typo — we truncate when performance begins to saturate. This is a choice we made to speed up experiments, but it is certainly possible to run environments for longer.
>
> **W4: We add repeated runs of experiments to characterize variance**.
> We repeated the open-ended experiments 3x for each domain. The open-ended environment is the least constrained so we expect the highest variance here. The overall improvement is higher than the variance in each case.
>
> |  | Multimodal (GQA) | MATH  | LiveCodeBench  |
> |---|---|---|---|
> | Before Teaching  | 44.18  | 15.78  | 16.50  |
> | Open-Ended (3 runs) | 53.25 $\\pm$ 1.97 | 21.55 $\\pm$ 1.42 | 18.55 $\\pm$ 0.27 |
>
> **Q1: How does the performance of the data generation agents change over longer interactions?**
> It differs by environment. In the MATH and LiveCodeBench environments, the performance saturates with increased training. In the GQA environment, the performance seems to continue increasing up to 56%, but becomes more unstable (fluctuations up and down).
>
> **Q2: Is the total training data fixed in each allocation, or does it vary dynamically?**
> We set a maximum budget for an experiment and terminate the experiment when the budget is exhausted or the student saturates, whichever happens earlier. It is up to the policy to decide how it wants to allocate the budget across skills and iterations. In the baseline policies, we leave this decision up to the LLM except for the skill-tree environment, where we allocate data uniformly across skills and subskills because it is a reasonable baseline.

---

> ### Author Response · Authors · 2024-11-22
>
> Thank you once again for your valuable feedback! We hope our response has addressed all of your questions and will allow you to revisit your score. We would be happy to engage further and address any further questions you might have in the remaining few days of the discussion period.

---

> ### Comment · Reviewer_rVo8 · 2024-11-23
>
> I thank the authors for the new experiments and clarifications.
>
> > This is not a typo — we truncate when performance begins to saturate. This is a choice we made to speed up experiments, but it is certainly possible to run environments for longer.
>
> That does not mean that the performance decreases. Decreases mean that the accuracy is dropping. Also, it is not clear in Figure 5 if the performance increase saturated.
>
> > We repeated the open-ended experiments 3x for each domain. The open-ended environment is the least constrained so we expect the highest variance here. The overall improvement is higher than the variance in each case.
>
> Why not include it in Figure 5?
>
> > It differs by environment. In the MATH and LiveCodeBench environments, the performance saturates with increased training. In the GQA environment, the performance seems to continue increasing up to 56%, but becomes more unstable (fluctuations up and down).
>
> Given this, why not include the full training progression in Figure 5 instead of truncating it? Providing more clarification on the decision to truncate would be helpful. Alternatively, adding an indicator on the figure to show where the truncation occurred and illustrating what the longer training progression would look like could address this.

---

> > ### Author Response · Authors · 2024-11-24
> > **Followup to Reviewer rVo8**
> >
> > Thanks for the great suggestions and continued engagement\!
> >
> > We've redone Fig. 5 on training dynamics in the style you've suggested:
> >
> > 1. We've run extended experiments instead of truncating them.
> > 2. We now show entire training curves to illustrate what the longer training progression would look like and add visual elements to show where the truncation occurred.
> > 3. We've added error bands based on our re-runs to characterize variance.
> > 4. We’ve rephrased Lines 460 \- 465 as per your suggestions.

---

> > > ### Comment · Reviewer_rVo8 · 2024-11-25
> > >
> > > Thank you for the additional experiments and explanation. I have updated my score accordingly.

---

> > > > ### Author Response · Authors · 2024-11-25
> > > >
> > > > Thank you, reviewer rVo8! We sincerely appreciate your thoughtfulness and engagement.

---

### Official Review · Reviewer_c5nB · 2024-11-04

**Soundness:** 4
**Presentation:** 3
**Contribution:** 3
**Rating:** 8
**Confidence:** 3

**Summary:**

This paper presents a modular system for automated data generation, designed to minimize the need for human annotations. The proposed approach employs a reinforcement learning-inspired methodology, decomposing the process into a sequence of action predictions (data generation policy) based on state information (feedback from model errors) in an iterative manner. The effectiveness of this approach is demonstrated through three diverse tasks, encompassing text, image, and code generation across different modalities.

**Strengths:**

This paper presents a novel and insightful perspective on the autonomous data generation problem, leveraging principles from reinforcement learning to conceptualize it as a sequential decision-making process. The authors provide a thorough explanation of this approach, the motivations behind and the underlying mechanics.

This paper proposed a modular framework/testbed that can be easily adapted to various tasks, showcasing its versatility and potential for widespread applicability. The authors demonstrate the effectiveness of their approach through experiments on 3 tasks of multiple modalities, including text, image, and code generation, yielding promising early results.

**Weaknesses:**

The experiment part should be conducted more thoroughly: specifically, creating a test set that incorporates newly generated data points from the data generation agent and reporting evaluation results for each retrained model over successive iterations would provide more comprehensive insights into the system's performance.

**Questions:**

In the Experiments section, the authors mention that the baseline student model should not have been heavily post-trained so that there are rooms for further improvements. However, it would be beneficial to provide additional evidence and details to support the claim that the student's performance is improved due to the added data points rather than insufficient training. For instance, the training protocol involved a fixed 10-epoch training period; it remains unclear whether the model had reached convergence within this timeframe or if the introduction of new data points accelerated convergence. Further clarification on this aspect would enhance the overall validity of the results.

Also the result would be more sound if more quantitative and qualitative results for skill discovery is reported in this paper.

---

> ### Author Response · Authors · 2024-11-20
> **Response to Reviewer c5nB**
>
> We’re glad you find DataEnvGym novel and insightful\!
>
> **W1: The student improves on generated test sets over successive iterations.**
> Following your suggestion, we conducted experiments with generated test sets. We summarize the results/findings below and have added them to Appendix D (L1173-1182) and Figure 12 (L1188,1199) in our revised PDF.
> For each setting, we show the performance of the student on test sets that incorporate newly generated data points over successive iterations.
> Concretely, we evaluate the performance of a student from iteration n on a test set created from data generated in iteration n+1 (unseen training data).
> This is only easily possible in the multimodal and MATH environments — the coding environment accuracy is determined by unit tests, which we do not currently generate.
> In all cases, the student improves on the generated test sets over successive iterations, and accuracy on the generated test set is higher in the last iteration than in the first.
>
> | Iteration | Accuracy (Generated Math Data) | Accuracy (Generated Multimodal Data) |
> |---|----|-----|
> | 0  | 29.25  | 45.52  |
> | 1  | 21.18  | 54.71  |
> | 2  | 29.41  | 53.85  |
> | 3  | 41.56  | 60.09  |
> | 4  | 41.03  | 57.66  |
> | 5  | 57.53  | N/A  |
> | 6  | 46.15  | N/A  |
> | 7  | 50  | N/A  |
> | 8  | 65.22  | N/A  |
> | 9  | 67.06  | N/A  |
>
> Note that the multimodal environments were only run for half the iterations of the mathematics environments.
>
> **Q1: The students' performance increases due to added data points rather than insufficient training.**
> To substantiate the claim that student performance is increased due to added data points rather than insufficient training, we take a subset of the data and increase the number of epochs such that the student receives a fraction of the added data, but an equivalent number of epochs as training on the full data.
> For example, if a student is normally trained for 10 epochs with 1000 generated training data, we take the data from the first data generation iteration (let’s say it contains 200 training data) and train an alternative student for $\\frac{1000}{200}\\times10=50$ epochs to isolate the effect of the generated training data vs the added training epochs.
> In each case, training with less data but for more epochs produces significantly smaller improvements than training with more data for fewer epochs, showing that *data* is responsible for increased performance rather than more training.
> In fact, extending training without additional data typically hurts performance — fresh data is essential. This highlights the importance of studying data generation as we do in our paper, as data generation is one of the few ways to get fresh data.
> We have added these results in Appendix E (Table 6\) in L1184-1223 in the revised PDF.
>
> |  | Data | Epochs | Accuracy (GQA) |
> |----|---|---|---|
> | Before Teaching  | \-  | \-  | 44.18  |
> | Less Data / Longer Training  | 20%  | 15  | 42.79  |
> | More Data / Standard Training | 100% | 3  | **47.9**  |
>
> |  | Data | Epochs | Accuracy (MATH) |
> |----|---|---|---|
> | Before Teaching  | \-  | \-  | 15.78  |
> | Less Data / Longer Training  | 10%  | 30  | 13.98  |
> | More Data / Standard Training | 100% | 3  | **23.44**  |
>
> |  | Data | Epochs | Accuracy (LiveCodeBench) |
> |----|---|---|---|
> | Before Teaching  | \-  | \-  | 16.5  |
> | Less Data / Longer Training  | 20%  | 15  | 15  |
> | More Data / Standard Training | 100% | 3  | **18.91**  |
>
> **Q2: We add more qualitative results for skill discovery.**
> Following your suggestion, we have added an additional figure showing a full list (Figure 10, L1115-1132 in the revised PDF)  of discovered skills for MATH, GQA, and LiveCodeBench in the SKILL-LIST environments in Appendix C. We have also added another figure showing qualitative examples of skill-errors that were fixed by training on synthetic data in Appendix C, highlighting the utility of skills in our framework. In summary, we now have 5  figures showing qualitative examples of skill discovery and one quantitative analysis of skill discovery in Appendix C.

---

> > ### Author Response · Authors · 2024-11-25
> > **Follow up to reviewer c5nB**
> >
> > Given that there is only one day remaining in the rebuttal period, **we wanted to gently check in whether our rebuttal addressed all your questions or we are happy to address any remaining questions.** We’ve added experiments to address your questions about (a) test sets that incorporate generated data (b) whether added data or training is responsible for performance increases and we also add more qualitative results on skill discovery. We hope that these additional results and answers will allow you to revisit your score — otherwise, we are happy to engage further!

---

> > ### Comment · Reviewer_c5nB · 2024-11-26
> >
> > Dear authors,
> >
> > Thank you for conducting the additional experiments and incorporating the results and findings. This addresses three of my major concerns:
> >
> > W1: Figure 12 demonstrates that students' performance improves when evaluated on test sets which incorporates the newly generated data points. Although the evaluation was only conducted on the multimodal and MATH environments, and not on the coding environment due to technical difficulties, I believe this set of experiments is well-designed and sound.
> >
> > Q1: Appendix E (Table 6) proves that the student performance is increased due to the added data rather than insufficient training initially. The result is valid and sound.
> >
> > Q2: The added figures (Figure 10), combined with the existing ones, provide a good coverage of both qualitative and quantitative results for skill discovery.
> >
> > Taking these into account, I have raised the score to 8.

---

> > > ### Author Response · Authors · 2024-11-27
> > >
> > > Thank you Reviewer c5nB! We're glad that you felt our additional experiments were well-designed + sound. We truly appreciate your effort in reviewing the paper and are grateful for your thoughtfulness in increasing your score.

---

### Official Review · Reviewer_VQ9Y · 2024-11-06

**Soundness:** 4
**Presentation:** 3
**Contribution:** 4
**Rating:** 8
**Confidence:** 4

**Summary:**

This paper introduces Gym environments for data synthesis, framing the problem as sequential decision-making. In these environments, actions correspond to data-generation plans, and states represent the performance summary of a student model. The paper implements environments for three tasks: visual question answering (VQA), math, and code generation. Each environment offers three state representations: open-ended, skill-list, and skill-tree. Additionally, it proposes an LLM-based policy for data generation. Experimental results demonstrate that the LLM can make strategically effective choices based on environment-state information.

**Strengths:**

- Tackle a timely and interesting problem.
- Provide the necessary infrastructure for the community to study the problem, opening up opportunities for future contributions.
- Consider various data generation strategies,
- Well-desgined experiments which demonstrate the effectiveness of the proposed approaches and conduct insightful analyses.

**Weaknesses:**

* The paper is currently dense and difficult to follow. The introduction includes excessive implementation details, which detract from providing a simple, high-level intuition. Using a specific task example to guide readers through the core concepts would make the paper more accessible.

* The paper focuses solely on the data generation plan rather than a full, end-to-end data generation process. It relies on a fixed, off-the-shelf data-generation engine that cannot be modified. The authors should admit this limitation and discuss potential strategies for overcoming it.

* The quality of the data-generation engine can impact both student performance and the data-generation plan itself. Current approaches do not take into account the data-generation engine capabilities in the design of the policy or the evaluation of the student. For instance, poor student performance might result from the engine producing low-quality data on a specific skill, which could prompt the policy to avoid querying the engine for that skill.

* The learning procedure can be resource-intensive. The authors should report the time, cost, and computing resources used for the experiments.

**Questions:**

- Is it possible to implement a random-policy baseline where you randomly chose a set of (naturally collected) datapoints from a data pool? The no-state baseline has flavor of this baseline but LLM-informed decisions could be biased.
- Is it possible to compare this approach with active learning, in which instead of doing data generation, you do data *selection* and ask models to generate only synthetic labels, but not synthetic inputs?

---

> ### Author Response · Authors · 2024-11-20
> **Response to Reviewer VQ9Y (Part 1/2)**
>
> Thank you for the quality feedback and for noticing our contributions to open-source infrastructure\!
>
> **W1: We have added a new Figure 6 in Appendix B (L864-884), guiding the reader through a concrete task example.**
> The figure walks a reader through a round of data generation for the multimodal task using GQA as an example.
>
> **W2-1: The data generation engine is swappable and not required for all domains**.
> The data generation engine is only fixed for the multimodal setting, where it relies on an off-the-shelf T2I model to generate images. For the code generation and math settings, the data generation policy directly produces the data in an end-to-end manner.
>
> **W2-2: What strategies exist for modifying the data generation engine?**
> Our framework easily allows updating the data generation agent (policy \+ engine) when using an open source LLM. For example, we could update the parameters of the data generation policy and data generation engine using experiences from multiple rounds of data generation through reinforcement learning.
>
> **W3: How can the teacher take into account the weaknesses of the data generation engine or itself?**
> We have designed DataEnvGym as an RL-style setting so the policy can learn over subsequent iterations what the data generation engine’s capabilities are. Our position is that the capabilities of the data generation engine should be discovered by the policy through a process of experimentation. The Skill-Tree environment explicitly provides a mechanism for this. Our framework supports policy learning of what the teaching capabilities of the agent are. For example, after allocating data for a skill and observing a lack of improvement, the policy can infer that the data generation engine has trouble with generating data for the skill and avoid data generation for that skill in subsequent iterations.
>
> **W4: Experiments can be run for fewer than \<$1 and under 10h on a single GPU.**
> On a single A6000, the total training time for our most computationally expensive setting (multimodal) is 6h, or about 1.5h/iteration. Most other settings are much faster. Environments are fully parallelizable using Ray and can be scaled up to multiple GPUs and even multiple nodes.
>
> We’ve added a table showing the token and time costs (Appendix B.4, Table 4, L918-931), which we summarize below. Additionally, we conduct experiments with a cheaper teacher, GPT-4o-mini, showing that it can be used as a cheaper alternative for GPT-4o. We’ve added these results in Appendix B.4 (Table 5, L972-986) of the revised PDF.
>
> | Domain  | Environment | Num Tokens | \$ Cost (GPT-4o-mini) | \$ Cost (GPT-4o) | GPU Minutes / Iteration |
> |---|---|---|---|---|---|
> | Math  | Open-Ended  | 173234  | 0.10  | 1.73  | 24  |
> | Math  | Skill-List  | 318528  | 0.19  | 3.19  | 24  |
> | Math  | Skill-Tree  | 355033  | 0.21  | 3.55  | 16  |
> | Coding  | Open-Ended  | 279304  | 0.17  | 2.79  | 16  |
> | Coding  | Skill-List  | 497787  | 0.30  | 4.98  | 16  |
> | Coding  | Skill-Tree  | 967610  | 0.58  | 9.68  | 16  |
> | Multimodal | Open-Ended  | 25073  | 0.02  | 0.25  | 37  |
> | Multimodal | Skill-List  | 82419  | 0.05  | 0.82  | 134  |
> | Multimodal | Skill-Tree  | 33991  | 0.02  | 0.34  | 78  |

---

> > ### Author Response · Authors · 2024-11-20
> > **Response to Reviewer VQ9Y (Part 2/2)**
> >
> > **Q1: We implement a random data selection baseline.**
> >
> > Data selection is not possible in general as many domains lack a data source from which to easily sample data (e.g., LiveCodeBench). Therefore, we implement it for MATH, as a standard training set is available. The random selection baseline cannot improve a student when sampling an equivalent amount of data as the data generation baseline. The results are shown below.
> >
> > We hypothesize that the random natural data selection baseline cannot improve a student like Gemma2-2B because easily accessible data pools (e.g., the training set for MATH) have already been exhausted by extensive LLM post-training \[C$\\S$4.2,D$\\S$4\] and so do not add new information.
> >
> > |  | Before Training | Random Data Selection | Data Generation (Without State) | Data Generation (With State) |
> > |---|---|---|----|----|
> > | MATH Accuracy | 15.78  | 15.26  | 19.78  | **23.44**  |
> >
> > **Q2: We implement a data selection agent.**
> >
> > We implement data selection using prototypicality scores \[A\] which are standard for active learning. Similar to the random selection baseline, it is hard to improve a well-post-trained LLM like Llama3 or Gemma2 by using readily available data pools — it is much easier to improve them using generated data. Even using the full training dataset cannot improve the student. This motivates our choice to tackle data generation rather than data selection. The training of open-source frontier models (Llama3, for example) includes significant post-training that subsumes publicly available data sources \[B$\\S$4.2, C$\\S$4\], making it hard to improve them with any amount of already existing data.
> >
> > | | Before Training | Data Selection (Prototypicality) | Full Training Dataset | Data Generation (Open-Ended) |
> > |---|---|-----|---|----|
> > | MATH Accuracy  | 15.78  | 16.01  | 15.18  | **23.44**  |
> >
> > \[A\] Sorscher et al., Beyond neural scaling laws: beating power law scaling via data pruning, NeurIPS 2022 Outstanding Paper Award
> > \[B\] Llama Team, AI @ Meta, The Llama 3 Herd of Models, arXiv 2024
> > \[C\] Gemma Team, Google Deepmind, Gemma 2: Improving Open Language Models at a Practical Size, arXiv 2024

---

> > > ### Comment · Reviewer_VQ9Y · 2024-11-25
> > > **Thank you**
> > >
> > > Thanks for the tremedous effort put into the response. It addressed most of my concerns so I raise the score to 8.

---

> > > > ### Author Response · Authors · 2024-11-26
> > > >
> > > > Thank you, reviewer VQ9Y! We truly appreciate your kind words and your effort in reviewing our work.

---

### Author Response · Authors · 2024-11-20
**General Response**

Reviewers believe we tackle “*a timely and interesting problem*” (VQ9Y) with a “*novel and insightful perspective on the autonomous data generation problem*” (c5nB), making a “*good contribution to automated data generation for model improvement*” (rVo8).

The potential impact of our work in making a challenging problem accessible is noted by several reviewers: “*necessary infrastructure for the community to study the problem*” (VQ9Y), that our “*structure makes the problem tractable*” (wuGW) and has “*potential for widespread applicability*” (c5nB).

We show that experiments can be run in half a day with limited compute resources (1x A6000) for under $1 of OpenAI API credits, making it an accessible testbed for developing data generation agents.

**We thank all reviewers for their valuable feedback and suggestions**. We have provided responses to all of the reviewer questions in the rebuttals in the individual responses and the revised PDF (updated text is in blue).

---

### Meta-Review · Area_Chair_eoLd · 2024-12-20

**Metareview:**

The paper frames the problem of automatic data generation (to improve a ML model) as a sequential decision making task, and provides Gym environments as well as LLM-based agents that are effective for them. The resulting datasets are shown to be effective for ML models in math reasoning, coding, and visual question answering domains.

All of the reviewers agreed that the paper is a solid contribution for ICLR. Reviewers praised the novelty and significance of the DataEnvGym, and anticipated follow-up works on data-generation agents.
The authors addressed the weaknesses identified by the reviewers very skillfully during the rebuttal.

**Additional Comments On Reviewer Discussion:**

Reviewers highlighted that there are missing experiment details (e.g. costs for running experiments) that the authors supplied during the rebuttal.
The authors also compared against active learning and random baselines for data generation/selection, and the proposed agents were substantially better across multiple domains.
Reviewers requested a more careful analysis of where the improvements were coming from (from the additional data, or from additional training) and the authors ran a thoughtful experiment to check.
All of these findings during the rebuttal period substantially strengthened the paper.

---

### Decision · Program_Chairs · 2025-01-22

Accept (Spotlight)